# Computational Doob's $h$-transforms for Online Filtering of Discretely Observed Diffusions

## Abstract

This paper is concerned with online filtering of discretely observed nonlinear diffu-
sion processes. Our approach is based on the fully adapted auxiliary particle filter
which involves Doob's $h$-transforms that are typically intractable. We propose a
computational framework to approximate these $h$-transforms by solving the under-
lying backward Kolmogorov equations using nonlinear Feynman-Kac formulas.
The methodology allows one to train a locally optimal particle filter prior to the
data-assimilation procedure. Numerical experiments illustrate that the proposed
approach can be orders of magnitude more efficient than the bootstrap particle fil-
ter in the regime of highly informative observations and when the observations are
extreme under the model.

## 1 Introduction

Diffusion processes are fundamental tools in applied mathematics, statistics and machine learning.
This rich class of continuous-time models have been used to model real-world phenomena in disci-
plines as varied as life-sciences, engineering, economics and finance. However, working with dif-
fusions can be challenging as its transition densities are only tractable in simple and specific cases
such as (geometric) Brownian motions, Ornstein-Uhlenbeck (OU) processes and Cox-Ingersoll-Ross
processes. This difficulty has hindered the use of standard methodologies for inference and data-
assimilation of models driven by diffusion processes. Specialized methodologies have been devel-
oped to circumvent or mitigate these issues [35, 4, 3, 14, 13, 5, 37].

Consider a time-homogeneous multivariate diffusion process $d\mathbf{X}_t = \mu(\mathbf{X}_t)\,dt + \sigma(\mathbf{X}_t)\,d\mathbf{B}_t$ that
is discretely observed at regular intervals. Noisy observations $\mathbf{y}_k$ of the latent process $\mathbf{X}_{t_k}$ is
collected at time $t_k \equiv k\,\mathrm{T}$ for $k \geq 1$. We consider the online filtering problem which consists in
estimating the conditional laws $\pi_k(d\mathbf{x}) = \mathbb{P}(\mathbf{X}_{t_k} \in d\mathbf{x}|\mathbf{y}_1,\ldots,\mathbf{y}_k)$, i.e. the filtering distributions,
as observations are collected. We focus on the use of Particle Filters (PF) that approximate the
filtering distributions with a system of weighted particles. Although many previous works have
relied on the Bootstrap Particle Filter (BPF), which simulates particles from the diffusion process,
it can perform poorly in challenging scenarios as it fails to take the incoming observation $\mathbf{y}_k$ into
account. This issue can be partially tackled by relying on resampling at intermediate times between
observations [10, 31]. The goal of this article is to show that the (locally) optimal approach given by
the Fully Adapted Auxiliary Particle Filter (FA-APF) [33] can be implemented. This necessitates
simulating a conditioned diffusion process, which can be formulated as a control problem involving
an intractable Doob's $h$-transform [36, 8]. We propose the *Computational Doob's $h$-Transform*
(CDT) framework for efficiently approximating these quantities. The method relies on nonlinear
Feynman-Kac formulas for solving backward Kolmogorov equations simultaneously for all possible
observations. Importantly, this preprocessing step only needs to be performed once before starting the
online filtering procedure. Numerical experiments illustrate that the proposed approach can be orders

of magnitude more efficient than the BPF in the regime of highly informative observations or when the observations are extreme under the model. A PyTorch implementation to reproduce our numerical experiments is available at https://anonymous.4open.science/r/CompDoobTransform/.

**Notations.** For two matrices $A, B \in \mathbb{R}^{d,d}$, their Frobenius inner product is defined as $\langle A, B \rangle_{\mathrm{F}} = \sum_{i,j=1}^{d} A_{i,j} B_{i,j}$. The Euclidean inner product for $\mathbf{u}, \mathbf{v} \in \mathbb{R}^d$ is denoted as $\langle \mathbf{u}, \mathbf{v} \rangle = \sum_{i=1}^{d} u_i v_i$.

## 2 Background

### 2.1 Filtering of discretely observed diffusions

Consider an homogeneous diffusion process $\{\mathbf{X}_t\}_{t \geq 0}$ in $\mathcal{X} = \mathbb{R}^d$ with initial distribution $\rho_0(d\mathbf{x})$ and dynamics

$$d\mathbf{X}_t = \mu(\mathbf{X}_t) \, dt + \sigma(\mathbf{X}_t) \, d\mathbf{B}_t, \tag{1}$$

described by the drift and volatility functions $\mu : \mathbb{R}^d \to \mathbb{R}^d$ and $\sigma : \mathbb{R}^d \to \mathbb{R}^{d,d}$. We assume standard smoothness and growth conditions [27] for a unique strong solution of (1) to exist for all times. The associated semi-group of transition probabilities $p_s(d\widehat{\mathbf{x}} \mid \mathbf{x})$ satisfies $\mathbb{P}(\mathbf{X}_{t+s} \in A \mid \mathbf{X}_t = \mathbf{x}) = \int_A p_s(d\widehat{\mathbf{x}} \mid \mathbf{x})$ for any $s, t > 0$ and measurable $A \subset \mathcal{X}$. The process $\{\mathbf{B}_t\}_{t \geq 0}$ is a standard $\mathbb{R}^d$-valued Brownian motion. The diffusion process $\{\mathbf{X}_t\}_{t \geq 0}$ is discretely observed at time $t_k = k\mathrm{T}$, for $k \geq 1$, for some inter-observation time $\mathrm{T} > 0$. The $\mathcal{Y}$-valued observation $\mathbf{Y}_k \in \mathcal{Y}$ at time $t_k$ is modelled by the likelihood function $g : \mathcal{X} \times \mathcal{Y} \to \mathbb{R}_+$ in the sense that for any measurable $A \subset \mathcal{Y}$, we have $\mathbb{P}(\mathbf{Y}_k \in A \mid \mathbf{X}_{t_k} = \mathbf{x}_k) = \int_A g(\mathbf{x}_k, \mathbf{y}) \, d\mathbf{y}$ for some dominating measure $d\mathbf{y}$ on $\mathcal{Y}$. For a test function $\varphi : \mathcal{X} \to \mathbb{R}$, the generator of the diffusion process $\{\mathbf{X}_t\}_{t \geq 0}$ is given by

$$\mathcal{L}\varphi = \langle \mu, \nabla \varphi \rangle + \frac{1}{2} \langle \sigma \sigma^\top, \nabla^2 \varphi \rangle_{\mathrm{F}}. \tag{2}$$

This article is concerned with approximating the filtering distributions $\pi_k(d\mathbf{x}) = \mathbb{P}(\mathbf{X}_{t_k} \in d\mathbf{x} \mid \mathbf{y}_1, \ldots, \mathbf{y}_k)$. For notational convenience, we set $\pi_0(d\mathbf{x}) \equiv \rho_0(d\mathbf{x})$.

### 2.2 Particle filtering

Particle Filters (PF), also known as Sequential Monte Carlo methods, are a set of Monte Carlo algorithms that can be used to solve filtering problems (see [7] for a recent textbook on the topic). PFs evolve a set of $N \geq 1$ particles $\mathbf{x}_t^{1:N} = (\mathbf{x}_t^1, \ldots, \mathbf{x}_t^N) \in \mathcal{X}^N$ forward in time using a combination of *propagation* and *resampling* operations.

To initialize the PF, each initial particle $\mathbf{x}_0^j \in \mathcal{X}$ for $1 \leq j \leq N$ is sampled independently from the distribution $\rho_0(d\mathbf{x})$ so that $\pi_0(d\mathbf{x}) \approx N^{-1} \sum_{j=1}^{N} \delta(d\mathbf{x}; \mathbf{x}_0^j)$. Approximations of the filtering distribution $\pi_k$ for $k \geq 1$ are built recursively as follows. Given the Monte Carlo approximation of the filtering distribution at time $t_k$, $\pi_k(d\mathbf{x}) \approx N^{-1} \sum_{j=1}^{N} \delta(d\mathbf{x}; \mathbf{x}_{t_k}^j)$, the particles $\mathbf{x}_{t_k}^{1:N}$ are propagated independently forward in time by $\widehat{\mathbf{x}}_{t_{k+1}}^j \sim q_{k+1}(d\widehat{\mathbf{x}} \mid \mathbf{x}_{t_k}^j)$, using a Markov kernel $q_{k+1}(d\widehat{\mathbf{x}} \mid \mathbf{x})$ specified by the user. The BPF corresponds to the choice of Markov kernel $q_{k+1}^{\mathrm{BPF}}(d\widehat{\mathbf{x}} \mid \mathbf{x}) = \mathbb{P}(\mathbf{X}_{t_{k+1}} \in d\widehat{\mathbf{x}} \mid \mathbf{X}_{t_k} = \mathbf{x})$ while the FA-APF [33] corresponds to the choice

$$q_{k+1}^{\mathrm{FA\text{-}APF}}(d\widehat{\mathbf{x}} \mid \mathbf{x}) = \mathbb{P}(\mathbf{X}_{t_{k+1}} \in d\widehat{\mathbf{x}} \mid \mathbf{X}_{t_k} = \mathbf{x}, \mathbf{Y}_{k+1} = \mathbf{y}_{k+1}). \tag{3}$$

Each particle $\widehat{\mathbf{x}}_{t_{k+1}}^j$ is associated with a normalized weight $\overline{W}_{k+1}^j = W_{k+1}^j / \sum_{i=1}^{N} W_{k+1}^i$, where the unnormalized weights $W_{k+1}^j > 0$ are defined as

$$W_{k+1}^j = \frac{p_{\mathrm{T}}(d\widehat{\mathbf{x}}_{t_{k+1}}^j \mid \mathbf{x}_{t_k}^j)}{q_{k+1}(d\widehat{\mathbf{x}}_{t_{k+1}}^j \mid \mathbf{x}_{t_k}^j)} \, g(\widehat{\mathbf{x}}_{t_{k+1}}^j, \mathbf{y}_{k+1}). \tag{4}$$

The BPF and FA-APF correspond respectively to having

$$W_{k+1}^{j,\mathrm{BPF}} = g(\widehat{\mathbf{x}}_{t_{k+1}}^j, \mathbf{y}_{k+1}) \qquad \text{and} \qquad W_{k+1}^{j,\mathrm{FA\text{-}APF}} = \mathbb{E}[g(\mathbf{X}_{t_{k+1}}, \mathbf{y}_{k+1}) \mid \mathbf{X}_{t_k} = \mathbf{x}_{t_k}^j]. \tag{5}$$

The weights are such that $\pi_{k+1}(d\mathbf{x}) \approx \sum_{j=1}^{N} \overline{W}_{k+1}^j \delta(d\mathbf{x}; \mathbf{x}_{t_{k+1}}^j)$. The *resampling* step consists in defining a new set of particles $\mathbf{x}_{t_{k+1}}^{1:N}$ with $\mathbb{P}(\mathbf{x}_{t_{k+1}}^j = \widehat{\mathbf{x}}_{t_{k+1}}^i) = \overline{W}_{k+1}^i$. This resampling scheme ensures that the equally weighted set of particles $\mathbf{x}_{t_{k+1}}^{1:N}$ provides a Monte Carlo approximation of the filtering distribution at time $t_{k+1}$ in the sense that $\pi_{k+1}(d\mathbf{x}) \approx N^{-1} \sum_{j=1}^{N} \delta(d\mathbf{x}; \mathbf{x}_{t_{k+1}}^j)$. Note that the particles $\mathbf{x}_{t_{k+1}}^{1:N}$ do not need to be resampled independently given the set of propagated particles $\widehat{\mathbf{x}}_{t_{k+1}}^{1:N}$. We refer the reader to [15] for a recent discussion of resampling schemes within PFs and to [9] for a book-length treatment of the convergence properties of this class of Monte Carlo methods.

In most settings, the FA-APF [33] that minimizes a local variance criterion [12] leads to better performance when compared to the BPF. This gain in efficiency can be very substantial when the signal-to-noise ratio is high or when observations contain outliers under the model specification. Nevertheless, implementing FA-APF requires sampling from the conditioned transition probability in (3), which is typically not feasible in practice. We will show in the following that this can be achieved in our setting by simulating a conditioned diffusion. We note also that standard strategies to approximate the FA-APF do not apply to our setup as the latent state process evolves on a higher frequency relative to the observations.

## 2.3 Conditioned and controlled diffusions

As the diffusion process (1) is assumed to be time-homogeneous, it suffices to focus on the initial interval $[0, \mathrm{T}]$ and study the dynamics of the diffusion $\mathbf{X}_{[0,\mathrm{T}]} = \{\mathbf{X}_t\}_{t \in [0,\mathrm{T}]}$ conditioned upon the first observation $\mathbf{Y}_\mathrm{T} = \mathbf{y}$. The conditioned dynamics can also be described by a diffusion process. Contrarily to the original diffusion, the conditioned process is not time-homogeneous in general. The conditioned process is described by the same volatility function but with a different drift term that takes the future observation $\mathbf{Y}_\mathrm{T} = \mathbf{y}$ into account.

Before deriving the exact form of the conditioned diffusion in Section 2.4, this section describes a more general setting that will be of crucial importance in our proposed numerical scheme. For a *control* function $\mathbf{c} : \mathcal{X} \times \mathcal{Y} \times [0, \mathrm{T}] \to \mathbb{R}^d$ and a given observation $\mathbf{y} \in \mathcal{Y}$, consider the controlled diffusion process $\{\mathbf{X}_t^{\mathbf{c}, \mathbf{y}}\}_{t \in [0,\mathrm{T}]}$ satisfying

$$d\mathbf{X}_t^{\mathbf{c}, \mathbf{y}} = \underbrace{\mu(\mathbf{X}_t^{\mathbf{c}, \mathbf{y}}) \, dt + \sigma(\mathbf{X}_t^{\mathbf{c}, \mathbf{y}}) \, d\mathbf{B}_t}_{\text{(original dynamics)}} + \underbrace{[\sigma \, \mathbf{c}](\mathbf{X}_t^{\mathbf{c}, \mathbf{y}}, \mathbf{y}, t) \, dt}_{\text{(control drift term)}}. \tag{6}$$

The dynamics of the controlled diffusion is identical to the original diffusion, except for the additional drift term $[\sigma \, \mathbf{c}](\mathbf{x}, \mathbf{y}, t) \in \mathbb{R}^d$ described by the control function $\mathbf{c}$. For $\mathbf{y} \in \mathcal{Y}$ and a test function $\varphi : \mathcal{X} \to \mathbb{R}$, the generator of the controlled diffusion is given by

$$\mathcal{L}^{\mathbf{c}, \mathbf{y}, t} \varphi(\mathbf{x}) = \mathcal{L}\varphi(\mathbf{x}) + \langle [\sigma \mathbf{c}](\mathbf{x}, \mathbf{y}, t), \nabla\varphi(\mathbf{x}) \rangle. \tag{7}$$

Let $\mathbb{P}_{[0,\mathrm{T}]}$ and $\mathbb{P}_{[0,\mathrm{T}]}^{\mathbf{c}, \mathbf{y}}$ denote the probability measures on the space of continuous functions $C([0, \mathrm{T}], \mathbb{R}^d)$ generated by the original and controlled diffusions respectively. Under mild growth assumptions on the control $\mathbf{c}$, the two measures are equivalent and Girsanov's theorem [16] shows that

$$\frac{d\mathbb{P}_{[0,\mathrm{T}]}}{d\mathbb{P}_{[0,\mathrm{T}]}^{\mathbf{c}, \mathbf{y}}}(\mathbf{X}_{[0,\mathrm{T}]}) = \exp\left\{ -\frac{1}{2} \int_0^\mathrm{T} \|\mathbf{c}(\mathbf{X}_t, \mathbf{y}, t)\|^2 \, dt - \int_0^\mathrm{T} \langle \mathbf{c}(\mathbf{X}_t, \mathbf{y}, t), d\mathbf{B}_t \rangle \right\}. \tag{8}$$

Our main objective is to construct a control function $\mathbf{c}_\star : \mathcal{X} \times \mathcal{Y} \times [0, \mathrm{T}] \to \mathbb{R}^d$ so that, for any observation value $\mathbf{y} \in \mathcal{Y}$, the controlled diffusion $\mathbf{X}_{[0,\mathrm{T}]}^{\mathbf{c}_\star, \mathbf{y}}$ has the same dynamics as the original diffusion $\mathbf{X}_{[0,\mathrm{T}]}$ conditioned upon the observation $\mathbf{Y}_\mathrm{T} = \mathbf{y}$, i.e. for any measurable set $A \subset C([0, \mathrm{T}], \mathbb{R}^d)$, we have

$$\mathbb{P}\left( \mathbf{X}_{[0,\mathrm{T}]}^{\mathbf{c}_\star, \mathbf{y}} \in A \right) = \mathbb{E}\left[ \mathbf{1}(\mathbf{X}_{[0,\mathrm{T}]} \in A) \, g(\mathbf{X}_\mathrm{T}, \mathbf{y}) \right] / \mathbb{E}[g(\mathbf{X}_\mathrm{T}, \mathbf{y})]. \tag{9}$$

We will give an exact expression of this control in Section 2.4 and propose a numerical scheme to approximate it in Section 3.1.

## 2.4 Doob's $h$-transform

To simplify notation, we shall denote the conditioned process $\mathbf{X}_{[0,\mathrm{T}]} \mid (\mathbf{Y}_\mathrm{T} = \mathbf{y})$ as $\widehat{\mathbf{X}}_{[0,\mathrm{T}]}$. To describe its dynamics, we introduce the function

$$h(\mathbf{x}, \mathbf{y}, t) = \mathbb{E}[g(\mathbf{X}_\mathrm{T}, \mathbf{y}) \mid \mathbf{X}_t = \mathbf{x}] = \int_{\mathcal{X}} g(\mathbf{x}_\mathrm{T}, \mathbf{y}) \, p_{\mathrm{T}-t}(d\mathbf{x}_\mathrm{T} \mid \mathbf{x}) \tag{10}$$

which gives the probability of observing $\mathbf{Y}_T = \mathbf{y}$ when the diffusion has state $\mathbf{x} \in \mathcal{X}$ at time $t \in [0, \mathrm{T}]$. We recall that the likelihood function $g : \mathcal{X} \times \mathcal{Y} \to \mathbb{R}_+$ was defined in Section 2.1. The definition in (10) implies that the function $h : \mathcal{X} \times \mathcal{Y} \times [0, T] \to \mathbb{R}_+$ satisfies the backward Kolmogorov equation [27],

$$(\partial_t + \mathcal{L})h = 0, \tag{11}$$

with terminal condition $h(\mathbf{x}, \mathbf{y}, \mathrm{T}) = g(\mathbf{x}, \mathbf{y})$ for all $(\mathbf{x}, \mathbf{y}) \in \mathcal{X} \times \mathcal{Y}$. For $\varphi : \mathcal{X} \to \mathbb{R}$ and an infinitesimal increment $\delta > 0$, we have

$$\begin{aligned}
\mathbb{E}[\varphi(\widehat{\mathbf{X}}_{t+\delta}) | \widehat{\mathbf{X}}_t = \mathbf{x}] &= \mathbb{E}[\varphi(\mathbf{X}_{t+\delta}) \, g(\mathbf{X}_\mathrm{T}, \mathbf{y}) \mid \mathbf{X}_t = \mathbf{x}] \, / \, \mathbb{E}[g(\mathbf{X}_\mathrm{T}, \mathbf{y}) | \mathbf{X}_t = \mathbf{x}] \\
&= \mathbb{E}[\varphi(\mathbf{X}_{t+\delta}) \, h(\mathbf{X}_{t+\delta}, \mathbf{y}, t+\delta) \mid \mathbf{X}_t = \mathbf{x}] \, / \, h(\mathbf{x}, \mathbf{y}, t) \\
&= \varphi(\mathbf{x}) + \delta \left\{ \frac{\mathcal{L}[\varphi \, h]}{h} \right\}(\mathbf{x}, \mathbf{y}, t) + O(\delta^2).
\end{aligned} \tag{12}$$

Furthermore, since the function $h$ satisfies (11), some algebra shows that $\mathcal{L}[\varphi \, h]/h = \mathcal{L}\varphi + \langle \sigma\sigma^\top \nabla \log h, \nabla\varphi \rangle$. By taking $\delta \to 0$, this heuristic derivation shows that the generator of the conditioned diffusion equals $\mathcal{L}\varphi + \langle \sigma\sigma^\top \nabla \log h, \nabla\varphi \rangle$. Hence $\widehat{\mathbf{X}}_{[0,\mathrm{T}]}$ satisfies the dynamics of a controlled diffusion (6) with control function $\mathbf{c}_\star(\mathbf{x}, \mathbf{y}, t) = [\sigma^\top \nabla \log h](\mathbf{x}, \mathbf{y}, t)$. We refer readers to [36, 8] for a formal treatment of Doob's $h$-transform.

## 2.5 Nonlinear Feynman-Kac formula

Obtaining the control function $\mathbf{c}_\star(\mathbf{x}, \mathbf{y}, t) = [\sigma^\top \nabla \log h](\mathbf{x}, \mathbf{y}, t)$ by solving the backward Kolmogorov equation in (11) for each possible observation $\mathbf{y} \in \mathcal{Y}$ is computationally not feasible. Furthermore, when the dimensionality of the state-space $\mathcal{X}$ becomes larger, standard numerical methods for solving Partial Differential Equations (PDEs) such as Finite Differences or the Finite Element Method become impractical. For these reasons, we propose instead to approximate the control function $\mathbf{c}_\star$ with neural networks, and employ methods based on automatic differentiation and the nonlinear Feynman-Kac approach to solve semilinear PDEs [19, 20, 24, 17, 6, 22, 23, 1, 18].

As the non-negative function $h$ typically decays exponentially for large $\|\mathbf{x}\|$, it is computationally more stable to work on the logarithmic scale and approximate the *value* function $v(\mathbf{x}, \mathbf{y}, t) = -\log[h(\mathbf{x}, \mathbf{y}, t)]$. Using the fact that $h$ satisfies the PDE (11), the value function satisfies

$$(\partial_t + \mathcal{L})v = \frac{1}{2} \|\sigma^\top \nabla v\|^2, \quad v(\mathbf{x}, \mathbf{y}, \mathrm{T}) = -\log[g(\mathbf{x}, \mathbf{y})] \quad \text{for all} \quad (\mathbf{x}, \mathbf{y}) \in \mathcal{X} \times \mathcal{Y}. \tag{13}$$

Let $\{\mathbf{X}_t^{\mathbf{c},\mathbf{y}}\}_{t \in [0,\mathrm{T}]}$ be a controlled diffusion defined in Equation (6) with a given control function $\mathbf{c} : \mathcal{X} \times \mathcal{Y} \times [0, \mathrm{T}] \to \mathbb{R}^d$ and define the diffusion process $\{\mathrm{V}_t\}_{t \in [0,\mathrm{T}]}$ as $\mathrm{V}_t = v(\mathbf{X}_t^{\mathbf{c},\mathbf{y}}, \mathbf{y}, t)$. Itô Lemma shows that for any observation $\mathbf{Y}_\mathrm{T} = \mathbf{y}$ and $0 \le s \le \mathrm{T}$, we have

$$\mathrm{V}_\mathrm{T} = \mathrm{V}_s + \int_s^\mathrm{T} \left( \frac{1}{2} \|\mathbf{Z}_t\|^2 + \langle c, \mathbf{Z}_t \rangle \right) dt + \int_s^\mathrm{T} \langle \mathbf{Z}_t, d\mathbf{B}_t \rangle$$

with $\mathbf{Z}_t = [\sigma^\top \nabla v](\mathbf{X}_t^{\mathbf{c},\mathbf{y}}, \mathbf{y}, t)$ and $\mathrm{V}_\mathrm{T} = -\log[g(\mathbf{X}_\mathrm{T}^{\mathbf{c},\mathbf{y}}, \mathbf{y})]$. In summary, the pair of processes $(\mathrm{V}_t, \mathbf{Z}_t)$ are such that the following equation holds,

$$-\log[g(\mathbf{X}_\mathrm{T}^{\mathbf{c},\mathbf{y}}, \mathbf{y})] = \mathrm{V}_s + \int_s^\mathrm{T} \left\{ \frac{1}{2} \|\mathbf{Z}_t\|^2 + \langle \mathbf{c}, \mathbf{Z}_t \rangle \right\} dt + \int_s^\mathrm{T} \langle \mathbf{Z}_t, d\mathbf{B}_t \rangle. \tag{14}$$

Crucially, under mild growth and regularity assumptions on the drift and volatility function $\mu : \mathcal{X} \to \mathbb{R}^d$ and $\sigma : \mathcal{X} \to \mathbb{R}^{d,d}$, the pair of processes $(\mathrm{V}_t, \mathbf{Z}_t)$ is the unique solution to Equation (14) [28, 29, 30, 40]. This result can be used as a building block for designing Monte Carlo approximations of the solution to semilinear and fully nonlinear PDEs [18, 34, 21]

## 3 Method

### 3.1 Computational Doob's $h$-transform

As before, consider a diffusion $\{\mathbf{X}_t^{\mathbf{c},\mathbf{y}}\}_{t\in[0,\mathrm{T}]}$ controlled by a function $\mathbf{c} : \mathcal{X} \times \mathcal{Y} \times [0,\mathrm{T}] \to \mathbb{R}^d$ and driven by the standard Brownian motion $\{\mathbf{B}_t\}_{t\geq 0}$. Furthermore, for two functions $\mathrm{N}_0 : \mathcal{X} \times \mathcal{Y} \to \mathbb{R}$ and $\mathrm{N} : \mathcal{X} \times \mathcal{Y} \times [0,\mathrm{T}] \to \mathbb{R}^d$, consider the diffusion process $\{\mathrm{V}_t\}_{t\in[0,\mathrm{T}]}$ defined as

$$\mathrm{V}_t = \mathrm{V}_0 + \int_0^s \left\{ \frac{1}{2} \|\mathbf{Z}_t\|^2 + \langle \mathbf{c}(\mathbf{X}_t^{\mathbf{c},\mathbf{y}}, \mathbf{y}, t), \mathbf{Z}_t \rangle \right\} dt + \int_0^s \langle \mathbf{Z}_t, d\mathbf{B}_t \rangle, \tag{15}$$

where the initial condition $\mathrm{V}_0$ and the process $\{\mathbf{Z}_t\}_{t\in[0,\mathrm{T}]}$ are defined as

$$\mathrm{V}_0 = \mathrm{N}_0(\mathbf{X}_0^{\mathbf{c},\mathbf{y}}, \mathbf{y}) \qquad \text{and} \qquad \mathbf{Z}_t = \mathrm{N}(\mathbf{X}_t^{\mathbf{c},\mathbf{y}}, \mathbf{y}, t). \tag{16}$$

Importantly, we remind the reader that the two diffusion processes $\mathbf{X}_t^{\mathbf{c},\mathbf{y}}$ and $\mathrm{V}_t$ are driven by the same Brownian motion $\mathbf{B}_t$. The uniqueness result mentioned at the end of Section 2.5 implies that, if for any choice of initial condition $\mathbf{X}_0^{\mathbf{c},\mathbf{y}} \in \mathcal{X}$ and terminal observation $\mathbf{y} \in \mathcal{Y}$ the condition $\mathrm{V}_\mathrm{T} = -\log[g(\mathbf{X}_\mathrm{T}^{\mathbf{c},\mathbf{y}}, \mathbf{y})]$ is satisfied, then we have that for all $(\mathbf{x}, \mathbf{y}, t) \in \mathcal{X} \times \mathcal{Y} \times [0,\mathrm{T}]$

$$\mathrm{N}_0(\mathbf{x}, \mathbf{y}) = -\log h(\mathbf{x}, \mathbf{y}, 0) \qquad \text{and} \qquad \mathrm{N}(\mathbf{x}, \mathbf{y}, t) = -[\sigma^\top \nabla \log h](\mathbf{x}, \mathbf{y}, t). \tag{17}$$

In particular, the optimal control is given by $\mathbf{c}_\star(\mathbf{x}, \mathbf{y}, t) = -\mathrm{N}(\mathbf{x}, \mathbf{y}, t)$.

These remarks suggest parametrizing the functions $\mathrm{N}_0(\cdot, \cdot)$ and $\mathrm{N}(\cdot, \cdot, \cdot)$ by two neural networks with respective parameters $\theta_0 \in \Theta_0$ and $\theta \in \Theta$ while minimizing the loss function

$$\mathrm{L}(\theta_0, \theta; \mathbf{c}) = \mathbb{E}\left[ \left( \mathrm{V}_\mathrm{T} + \log[g(\mathbf{X}_\mathrm{T}^{\mathbf{c},\mathbf{Y}}, \mathbf{Y})] \right)^2 \right]. \tag{18}$$

The above expectation is with respect to the distribution of the Brownian motion $\{\mathbf{B}_t\}_{t\geq 0}$, the initial condition $\mathbf{X}_0^{\mathbf{c},\mathbf{Y}} \sim \eta_\mathbf{X}(d\mathbf{x})$ of the controlled diffusion, and the observation $\mathbf{Y} \sim \eta_\mathbf{Y}(d\mathbf{y})$ at time T. In practice, we will let the three sources of randomness be independent of each other. The spread of the distributions $\eta_\mathbf{X}$ and $\eta_\mathbf{Y}$ should be large enough to cover typical states under the filtering distributions $\pi_k, k \geq 1$ and future observations to be filtered respectively. Specific choices will be detailed for each application in Section 4. For offline problems, one could learn in a data-driven manner by selecting $\eta_\mathbf{Y}$ as the empirical distribution of actual observations. Furthermore, any control function $\mathbf{c} : \mathcal{X} \times \mathcal{Y} \times [0,\mathrm{T}] \to \mathbb{R}^d$ with mild growth and regularity assumptions can be employed within our methodology: specific choices are discussed at the end of this section.

**CDT algorithm.** The following outlines our training procedure to learn neural networks $\mathrm{N}_0$ and $\mathrm{N}$ that satisfy (17). To minimize the loss function (18), any stochastic gradient algorithm can be used with a user-specified *mini-batch* size of $J \geq 1$. The following steps are iterated until convergence.

1. Choose a control $\mathbf{c} : \mathcal{X} \times \mathcal{Y} \times [0,\mathrm{T}] \to \mathbb{R}^d$, possibly based on the current neural network parameters $(\theta_0, \theta) \in \Theta_0 \times \Theta$.

2. Simulate independent Brownian paths $\mathbf{B}_{[0,\mathrm{T}]}^j$, initial conditions $\mathbf{X}_0^j \sim \eta_\mathbf{X}(d\mathbf{x})$, and observations $\mathbf{Y}^j \sim \eta_\mathbf{Y}(d\mathbf{y})$ for $1 \leq j \leq J$.

3. Generate the controlled trajectories: the $j$-th sample path $\mathbf{X}_{[0,\mathrm{T}]}^j$ is obtained by forward integration of the controlled dynamics in Equation (6) with initial condition $\mathbf{X}_0^j$, control $\mathbf{c}(\cdot, \mathbf{Y}^j, \cdot)$, and the Brownian path $\mathbf{B}_{[0,\mathrm{T}]}^j$.

4. Generate the value trajectories: the $j$-th sample path $\mathrm{V}_{[0,\mathrm{T}]}^j$ is obtained by forward integration of the dynamics in Equation (15)–(16) with the Brownian path $\mathbf{B}_{[0,\mathrm{T}]}^j$ and the current neural network parameters $(\theta_0, \theta) \in \Theta_0 \times \Theta$.

5. Construct a Monte Carlo estimate of the loss function (18):

$$\widehat{\mathrm{L}} = J^{-1} \sum_{j=1}^J (\mathrm{V}_\mathrm{T}^j + \log[g(\mathbf{X}_\mathrm{T}^j, \mathbf{Y}^j)])^2 \tag{19}$$

180    6. Use automatic differentiation to compute $\partial_{\theta_0} \widehat{L}$ and $\partial_\theta \widehat{L}$ and update the parameters $(\theta_0, \theta)$.

181    Importantly, if the control function $\mathbf{c}$ in *Step:1* does depend on the current parameters $(\theta_0, \theta)$, the
182    gradient operations executed in *Step:6* should not be propagated through the control function $\mathbf{c}$. A
183    standard `stop-gradient` operation available in most popular automatic differentiation frameworks
184    can be used for this purpose.

185    **Time-discretization of diffusions.**    For clarity of exposition, we have described our algorithm in
186    continuous-time. In practice, one would have to discretize these diffusion processes, which is entirely
187    straightforward. Although any numerical integrator could potentially be considered, the experiments
188    in Section 4 employed the standard Euler-Maruyama scheme [25].

189    **Parametrizations of functions $N_0$ and $N$.**    In all numerical experiments presented in Section 4, the
190    functions $N_0$ and $N$ are parametrized with fully-connected neural networks with two hidden layers
191    and the Leaky ReLU activation function except in the last layer. Future work could explore other
192    neural network architectures for our setting.

193    **Choice of controlled dynamics.**    In challenging scenarios where observations are highly informative
194    and/or extreme under the model, choosing a good control function to implement *Step:1* of the
195    proposed algorithm can be crucial. We focus on two possible implementations:

196    - **CDT static scheme:** a simple (and naive) choice is not using any control, i.e. $\mathbf{c}(\mathbf{x}, \mathbf{y}, t) \equiv$
197      $0 \in \mathbb{R}^d$ for all $(\mathbf{x}, \mathbf{y}, t) \in \mathcal{X} \times \mathcal{Y} \times [0, \mathrm{T}]$.

198    - **CDT iterative scheme:** use the current approximation of the optimal control $\mathbf{c}_\star$ described
199      by the parameters $(\theta_0, \theta) \in \Theta_0 \times \Theta$. This corresponds to setting $\mathbf{c}(\mathbf{x}, \mathbf{y}, t) = -N(\mathbf{x}, \mathbf{y}, t)$.

200    While using a *static control* approach can perform reasonably well in some situations, our results in
201    Section 4 suggest that the *iterative control* procedure is a more reliable strategy. This is consistent
202    with findings in the stochastic optimal control literature [38, 32]. This choice of control function
203    drives the forward process $\mathbf{X}_t^{\mathbf{c},\mathbf{y}}$ to regions of the state-space where the likelihood function is large and
204    helps mitigate convergence and stability issues. Furthermore, Section 4 reports that (at convergence),
205    the solutions $N_0$ and $N$ can be significantly different. The *iterative control* procedure leads to more
206    accurate solutions and, ultimately, better performance when used for online filtering.

## 3.2 Online filtering

208    Before performing online filtering, we first run the CDT algorithm described in Section 3.1 to construct
209    an approximation of the optimal control $\mathbf{c}_\star(\mathbf{x}, \mathbf{y}, t) = [\sigma^\top \nabla \log h](\mathbf{x}, \mathbf{y}, t)$. For concreteness, denote
210    by $\widehat{\mathbf{c}} : \mathcal{X} \times \mathcal{Y} \times [0, \mathrm{T}] \to \mathbb{R}^d$ the resulting approximate control, i.e. $\widehat{\mathbf{c}}(\mathbf{x}, \mathbf{y}, t) = -N(\mathbf{x}, \mathbf{y}, t)$ where
211    $N(\cdot, \cdot, \cdot)$ is parametrized by the final parameter $\theta \in \Theta$. Similarly, denote by $\widehat{V}_0 : \mathcal{X} \times \mathcal{Y} \to \mathbb{R}$ the
212    approximation of the initial value function $v(\mathbf{x}, \mathbf{y}, 0) = -\log h(\mathbf{x}, \mathbf{y}, 0)$, i.e. $\widehat{V}_0(\mathbf{x}, \mathbf{y}) = N_0(\mathbf{x}, \mathbf{y})$
213    where $N_0(\cdot, \cdot)$ is parametrized by the final parameter $\theta_0 \in \Theta_0$.

214    For implementing online filtering with $N \geq 1$ particles, consider a current approximation $\pi_k(d\mathbf{x}) =$
215    $N^{-1} \sum_{j=1}^N \delta(d\mathbf{x}; \mathbf{x}_{t_k}^j)$ of the filtering distributions at time $t_k \geq 0$. Given the future observation
216    $\mathbf{Y}_{k+1} = \mathbf{y}_{k+1}$, the particles $\mathbf{x}_{t_k}^{1:N}$ are then propagated forward by exploiting the approximately
217    optimal control $(\mathbf{x}, t) \mapsto \widehat{\mathbf{c}}(\mathbf{x}, \mathbf{y}_{k+1}, t - t_k)$. In particular, $\widehat{\mathbf{x}}_{t_{k+1}}^j$ is obtained by setting $\widehat{\mathbf{x}}_{t_{k+1}}^j = \widehat{\mathbf{X}}_{t_{k+1}}^j$
218    where $\{\widehat{\mathbf{X}}_t^j\}_{t \in [t_k, t_{k+1}]}$ follows the controlled diffusion

$$d\widehat{\mathbf{X}}_t^j = \underbrace{\mu(\widehat{\mathbf{X}}_t^j)\, dt + \sigma(\widehat{\mathbf{X}}_t^j)\, d\mathbf{B}_t^j}_{\text{(original dynamics)}} + \underbrace{[\sigma \widehat{\mathbf{c}}](\widehat{\mathbf{X}}_t^j, \mathbf{y}_{k+1}, t - t_k)\, dt}_{\text{(approximately optimal control)}} \tag{20}$$

219    initialized at $\widehat{\mathbf{X}}_{t_k}^j = \mathbf{x}_{t_k}^j$. Each propagated particle $\widehat{\mathbf{x}}_{t_{k+1}}^j$ is associated with a normalized
220    weight $\overline{W}_{k+1}^j = W_{k+1}^j / \sum_{i=1}^N W_{k+1}^i$ where $W_{k+1}^j = (d\mathbb{P}_{[t_k, t_{k+1}]} / d\mathbb{P}_{[t_k, t_{k+1}]}^{\widehat{\mathbf{c}}, \mathbf{y}_{k+1}})(\widehat{\mathbf{X}}_{[t_k, t_{k+1}]}^j) \times$
221    $g(\widehat{\mathbf{x}}_{t_{k+1}}^j, \mathbf{y}_{k+1})$. We recall that the probability measures $\mathbb{P}_{[t_k, t_{k+1}]}$ and $\mathbb{P}_{[t_k, t_{k+1}]}^{\widehat{\mathbf{c}}, \mathbf{y}_{k+1}}$ correspond to the
222    original and controlled diffusions on the interval $[t_k, t_{k+1}]$. Girsanov's theorem, as described in Equa-
223    tion (8), implies that

$$W_{k+1}^j = \exp\left\{ -\frac{1}{2} \int_{t_k}^{t_{k+1}} \|\mathbf{Z}_t^j\|^2\, dt + \int_{t_k}^{t_{k+1}} \langle \mathbf{Z}_t^j, d\mathbf{B}_t^j \rangle + \log g(\mathbf{x}_{t_{k+1}}^j, \mathbf{y}_{k+1}) \right\}$$

where $\mathbf{Z}_t^j = -\widehat{\mathbf{c}}(\widehat{\mathbf{X}}_t^j, \mathbf{y}_{k+1}, t - t_k)$. Similarly to Equation (15), consider the diffusion process $\{\mathbf{V}_t^j\}_{t \in [t_k, t_{k+1}]}$ defined by the dynamics $d\mathbf{V}_t^j = -\frac{1}{2}\|\mathbf{Z}_t^j\|^2 \, dt + \langle \mathbf{Z}_t^j, d\mathbf{B}_t^j\rangle$ with initialization at $\mathbf{V}_{t_k}^j = \widehat{\mathbf{V}}_0(\mathbf{x}_{t_k}^j, \mathbf{y}_{k+1})$. Therefore the weight can be re-written as

$$W_{k+1}^j = \exp\left\{\underbrace{\mathbf{V}_{t_{k+1}}^j + \log g(\mathbf{x}_{t_{k+1}}^j, \mathbf{y}_{k+1})}_{\approx 0}\right\} \exp\left\{-\widehat{\mathbf{V}}_0(\mathbf{x}_{t_k}^j, \mathbf{y}_{k+1})\right\}, \tag{21}$$

and computed by numerically integrating the process $\{\mathbf{V}_t^j\}_{t \in [t_k, t_{k+1}]}$. Given the definition of the loss function in (18), we can expect the term within the first exponential to be close to zero. In the ideal case where $\widehat{\mathbf{c}}(\mathbf{x}, \mathbf{y}, t) \equiv \mathbf{c}_\star(\mathbf{x}, \mathbf{y}, t)$ and $\widehat{\mathbf{V}}_0(\mathbf{x}, \mathbf{y}) \equiv -\log h(\mathbf{x}, \mathbf{y}, 0)$, one recovers the exact AF-APF weights in (5). Once the unnormalized weights (21) are computed, the resampling steps are identical to those described in Section 2.2 for a standard PF. For practical implementations, all the processes involved in the proposed methodology can be straightforwardly time-discretized. To distinguish between CDT learning with static or iterative control, we shall refer to the resulting approximation of FA-APF as Static-APF and Iterative-APF respectively.

# 4   Experiments

This section presents numerical results obtained on three models. All experiments employed 2000 iterations of the Adam optimizer with a learning rate of $0.01$ and a mini-batch size of 1000 sample paths with 10 different observations. Training times took around one to two minutes on a standard CPU. We note that this compute time is marginal compared to the cost of running filters with many particles and/or to assimilate large number of observations. Moreover, we can also benefit from the use of hardware accelerators. We set the inter-observation time as $T = 1$ and employed the Euler-Maruyama integrator with a stepsize of $0.02$ for all examples. Our results are not sensitive to the choice of $T$ and discretization stepsize if it is sufficiently small. We examined the performance of each particle filter by computing its effective sample size (ESS) averaged over observation times and independent repetitions, the evidence lower bound (ELBO) $\mathbb{E}[\log \widehat{p}(\mathbf{y}_1, \ldots, \mathbf{y}_K)]$, and the variance $\mathrm{Var}[\log \widehat{p}(\mathbf{y}_1, \ldots, \mathbf{y}_K)]$, where $\widehat{p}(\mathbf{y}_1, \ldots, \mathbf{y}_K)$ denotes its unbiased estimator of the marginal likelihood of the time-discretized filter $p(\mathbf{y}_1, \ldots, \mathbf{y}_K)$. When testing particle filters with varying number of observations $K$, we increased the number of particles linearly with $K$ to keep marginal likelihood estimators stable [2].

## 4.1   Ornstein-Uhlenbeck model

We considered an Ornstein-Uhlenbeck process given by (1) with $\mu(\mathbf{x}) = -\mathbf{x}$, $\sigma(\mathbf{x}) = 1$ and the Gaussian observation model $g(\mathbf{x}, \mathbf{y}) = \mathcal{N}(\mathbf{y}; \mathbf{x}, \sigma_{\mathbf{Y}}^2)$. We chose $\eta_{\mathbf{X}} = \mathcal{N}(0, 1/2)$ as the stationary distribution and $\eta_{\mathbf{Y}} = \mathcal{N}(0, 1/2 + \sigma_{\mathbf{Y}}^2)$ as the implied distribution of the observation when training neural networks with the CDT iterative scheme. We took different values of $\sigma_{\mathbf{Y}} \in \{0.125, 0.25, 0.5, 1.0\}$ to vary the informativeness of observations. Analytically tractability in this example allows us to visualize the quality of our neural network approximations in Figure 1 and consider two idealized particle filters, namely an APF with exact networks (Exact-APF) and the FA-APF. Comparing our proposed Iterative-APF to Exact-APF and FA-APF enables us to distinguish between neural network approximation errors and time-discretization errors. We note that all PFs except the FA-APF involve time-discretization.

Columns 1 to 4 of Figure 2 summarize our numerical findings when filtering simulated observations from the model. We see that the performance of BPF deteriorates as the observations become more informative, which is to be expected. Furthermore, when $\sigma_{\mathbf{Y}}$ is small, the impact of our neural network approximation and time-discretization becomes more noticeable. For the values of $\sigma_{\mathbf{Y}}$ and the number of observations $K$ considered, we obtained around an order of magnitude gain in efficiency over BPF. From Column 5, we note that these gains become very substantial when we filter $K = 100$ observations that are simulated with observation noise that are several standard deviations larger than $\sigma_{\mathbf{Y}} = 0.25$ under the model specification. In particular, while the ELBO of BPF diverges as we increase the degree of noise in the simulated observations, the ELBO of Iterative-APF remains stable.

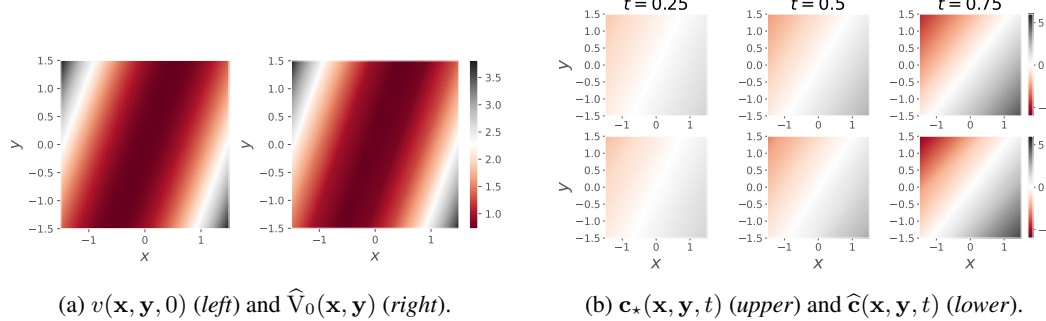

(a) $v(\mathbf{x}, \mathbf{y}, 0)$ (*left*) and $\widehat{V}_0(\mathbf{x}, \mathbf{y})$ (*right*).

(b) $\mathbf{c}_\star(\mathbf{x}, \mathbf{y}, t)$ (*upper*) and $\widehat{\mathbf{c}}(\mathbf{x}, \mathbf{y}, t)$ (*lower*).

Figure 1: Neural network approximations for Ornstein-Uhlenbeck model with $\sigma_{\mathbf{Y}} = 0.5$.

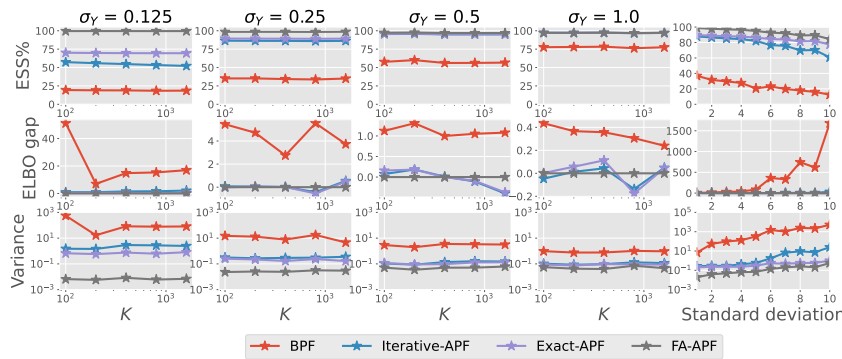

Figure 2: Results for Ornstein-Uhlenbeck model based on 100 independent repetitions of each PF. The ELBO gap in the second row is relative to FA-APF.

## 4.2 Logistic diffusion model

Next we consider a logistic diffusion process [11, 26] to model the dynamics of a population size $\{\mathbf{P}_t\}_{t \geq 0}$, defined by

$$d\mathbf{P}_t = (\theta_3^2/2 + \theta_1 - \theta_2 \mathbf{P}_t)\mathbf{P}_t \, dt + \theta_3 \mathbf{P}_t \, d\mathbf{B}_t. \tag{22}$$

We apply the Lamperti transformation $\mathbf{X}_t = \log(\mathbf{P}_t)/\theta_3$ and work with the process $\{\mathbf{X}_t\}_{t \geq 0}$ that satisfies (1) with $\mu(\mathbf{x}) = \theta_1/\theta_3 - (\theta_2/\theta_3) \exp(\theta_3 \mathbf{x})$ and $\sigma(\mathbf{x}) = 1$. Following [26], we adopt a negative binomial observation model $g(\mathbf{x}, \mathbf{y}) = \mathcal{NB}(\mathbf{y}; \theta_4, \exp(\theta_3 \mathbf{x}))$ for counts $\mathbf{y} \in \mathbb{N}_0$ with dispersion $\theta_4 > 0$ and mean $\exp(\theta_3 \mathbf{x})$. We set $(\theta_1, \theta_2, \theta_3, \theta_4)$ as the parameter estimates obtained in [26]. Noting that (22) admits a Gamma distribution with shape parameter $2(\theta_3^2/2 + \theta_1)/\theta_3^2 - 1$ and rate parameter $2\theta_2/\theta_3^2$ as stationary distribution [11], we select $\eta_{\mathbf{X}}$ as the push-forward under the Lamperti transformation and $\eta_{\mathbf{Y}}$ as the implied distribution of the observation when training neural networks under both static and iterative CDT schemes. To induce varying levels of informative observations, we considered $\theta_4 \in \{1.069, 4.303, 17.631, 78.161\}$.

Figure 3 displays our filtering results for various number of simulated observations from the model (Columns 1 to 4) and for $K = 100$ observations that are simulated with an observation model with several standard deviations larger than $\theta_4 = 17.631$ under the model specification (Column 5). In the latter setup, we solved for different values of $\theta_4$ in the negative binomial observation model to induce larger standard deviations. The behaviour of BPF and Iterative-APF is similar to the previous example as the observations become more informative with larger values of $\theta_4$. Iterative-APF outperforms both BPF and Static-APF over all combinations of $\theta_4$ and $K$ considered, and also when filtering observations that are increasingly extreme under the model. We note also that the APFs trained using the CDT static scheme can sometimes give unstable results, particularly in challenging scenarios.

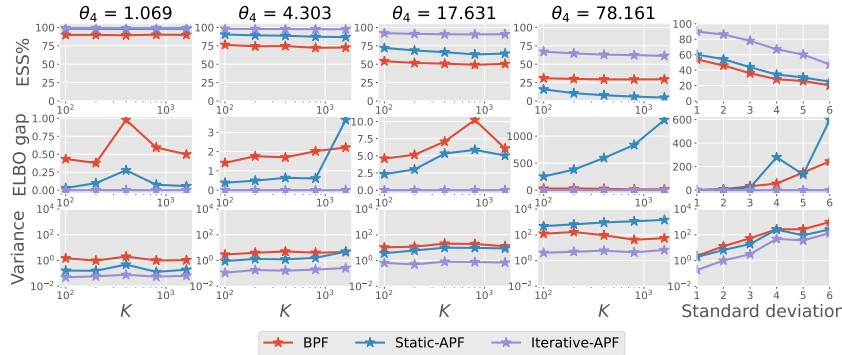

Figure 3: Results for logistic diffusion model based on 100 independent repetitions of each PF. The ELBO gap in the second row is relative to Iterative-APF.

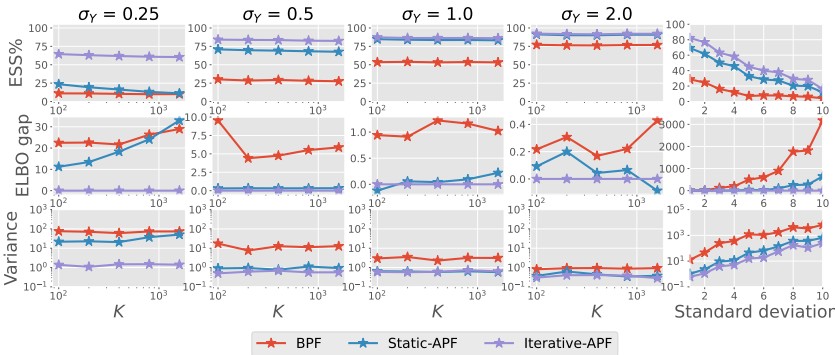

Figure 4: Results for cell model based on 100 independent repetitions of each PF. The ELBO gap in the second row is relative to Iterative-APF.

### 4.3 Cell model

Lastly, we examine a cell differentiation and development model from [39]. Cellular expression levels $\mathbf{X}_t = (\mathbf{X}_{t,1}, \mathbf{X}_{t,2})$ of two genes are modelled by (1) with

$$\mu(\mathbf{x}) = \begin{pmatrix} \mathbf{x}_1^4/(2^{-4} + \mathbf{x}_1^4) + 2^{-4}/(2^{-4} + \mathbf{x}_2^4) - \mathbf{x}_1 \\ \mathbf{x}_2^4/(2^{-4} + \mathbf{x}_2^4) + 2^{-4}/(2^{-4} + \mathbf{x}_1^4) - \mathbf{x}_2 \end{pmatrix} \tag{23}$$

and $\sigma(\mathbf{x}) = \sqrt{0.1} I_d$. The terms in (23) describe self-activation, mutual inhibition and inactivation respectively, and the volatility captures intrinsic and external fluctuations. We initialize the diffusion process from the undifferentiated state of $\mathbf{X}_0 = (1, 1)$ and consider the Gaussian observation model $g(\mathbf{x}, \mathbf{y}) = \mathcal{N}(\mathbf{y}; \mathbf{x}, \sigma_{\mathbf{Y}}^2 I_2)$. To train neural networks under both static and iterative CDT schemes, we selected $\eta_{\mathbf{X}}$ and $\eta_{\mathbf{Y}}$ as the empirical distributions obtained by simulating states and observations from the model for 2000 time units.

Figure 4 illustrates our numerical results for various number of observations $K$ and $\sigma_{\mathbf{Y}} \in \{0.25, 0.5, 1.0, 2.0\}$. It shows that Iterative-APF offers significant gains over BPF and Static-APF when filtering observations that are informative (see Columns 1 to 4) and highly extreme under the model specification of $\sigma_{\mathbf{Y}} = 0.5$ (see Column 5). In this example, Static-APF did not exhibit any unstable behaviour and its performance lies somewhere in between BPF and Iterative-APF.

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
