# OpenReview forum: "Computational Doob h-transforms for Online Filtering of Discretely Observed Diffusions"
_NeurIPS.cc/2022/Conference — NeurIPS 2022 Submitted_

### Official Review · Reviewer_AjSG · 2022-06-22

**Rating:** 5
**Confidence:** 4
**Soundness:** 3 good
**Presentation:** 3 good
**Contribution:** 2 fair

**Summary:**

This paper develops an approximation to the locally optimal particle filter in discretely observed diffusion processes under the assumption of a constant measurement rate.

**Questions:**


Q1 - The difference between CDT static scheme and bootstrap proposals needs to be elucidated. Choosing c = 0 yield the original SDE dynamics, i.e. bootstrap proposals, no?

**Limitations:**

Its fine.

**Strengths And Weaknesses:**

S1 - The method compares favourably to baseline solutions.

W1 - Experiments are not entirely convincing. More specifically, only low-dimensional examples re considered (d <= 2), which undermines the rationale for their method in that we are not in a high-dimensional regime where numerical  solutions of PDEs become infeasible (paragraph 1 of sec 2.5).

W2 - The method relies on fixed SDE dynamics so is not applicable to parameter inference.

W3 - The method relies on constant sample rate, which is many times infeasible in practice due to e.g. missing measurements.

---

> ### Author Response · Authors · 2022-08-02
> **Response to Reviewer AjSG [1 / 2]**
>
> We thank the reviewer for his/she comments. We give a point-by-point response to these comments below.
> ***
> 1. > Experiments are not entirely convincing. More specifically, only low-dimensional examples re considered $(d \leq 2)$, which undermines the rationale for their method in that we are not in a high-dimensional regime where numerical solutions of PDEs become infeasible (paragraph 1 of sec 2.5).
>
> Thanks for this comment. While applicability to higher-dimensional problems is indeed important, we would like to point out that online nonlinear filtering for diffusions is known to be already a hard problem in low dimensions. In low dimensions, even though one can run classical PDE solvers (e.g. finite difference and finite element methods), it is computationally impractical to use this for online filtering as one would require many solver calls (i.e. one call per observation). Hence we believe that our proposed approach based on a neural network solver is interesting even for low-dimensional problems.
>
> To illustrate the applicability of our approach to higher-dimensional settings where it is computationally infeasible to consider classical PDE solvers, we have added a numerical study of how our proposed and competing methods behave as the dimensionality increases. Here are the numerical results (with a link to our GitHub repository).
>
> [Impact of increasing dimension $d$](https://anonymous.4open.science/r/CompDoobTransform/figures/OU_dim_results.pdf)
>
> Here we have chosen to perform this study on the Ornstein-Uhlenbeck model as analytical tractability in this example allows us to carefully benchmark our approximations of the fully adapted auxiliary particle filter (FA-APF). Moreover, this linear Gaussian setting is still challenging for many methods that do not exploit the model structure, and the ideal FA-APF itself suffers from the curse of dimensionality [SBBA2008, SBM2015].
> ***
> 2. > The method relies on fixed SDE dynamics so is not applicable to parameter inference.
>
> Our paper is concerned with the problem of online filtering for (a given) diffusion, which is the setup considered by many works in this literature (e.g. [FPR2008, FPRS2010]). We note that this is already a formidable problem in the regime of informative or extreme observations, or in state-spaces with moderately high dimensions.
>
> Our proposed CDT algorithm can in fact be easily extended to incorporate and learn parameter dependence in the neural networks. We did not pursue this avenue as it is more sensible to consider parameter inference in an offline setting by approximating the smoothing distribution which conditions all observations (instead of the filtering distributions which only conditions on the data sequentially, in an online manner). We view this direction as distinct from the online filtering problem and outside the scope of the current manuscript (to be considered in future work). Following the reviewer's comment, we have added some discussion about parameter inference and the smoothing problem in the updated manuscript.
> ***
> 3. > The method relies on constant sample rate, which is many times infeasible in practice due to e.g. missing measurements.
>
> Thank you for raising this practical point. While we do not think our assumption of constant observation frequency is particularly restrictive in practice, i.e. many types of data are collected at regular time intervals, we do agree that it is worth extending our approach to deal with the setting of irregular observation frequencies.
>
> Due to the nature of the backward Kolmogorov equation, this is reasonably straightforward as long as the inter-observation time is upper bounded. For this purpose, one can for example learn the value function $v(x,y,t)$ satisfying Equation (13) as a function of the current state $x$, the next observation $y$, and the time to the next observation $T-t$. The control $c(x,y,t)=(\sigma^\top\nabla\log h)(x,y,t)$ is then obtained using automatic differentiation. This type of parametrization has already been employed in the backward SDE community for solving semi-linear PDEs [CMN2019]. We have added a few explanatory sentences about irregular observation frequencies in the main text.
> ***

---

> ### Author Response · Authors · 2022-08-02
> **Response to Reviewer AjSG [2 / 2]**
>
> 4. > The difference between CDT static scheme and bootstrap proposals needs to be elucidated. Choosing $c = 0$ yield the original SDE dynamics, i.e. bootstrap proposals, no?
>
> The CDT static scheme involves training the neural networks approximating the initial value and control functions by simulating sample paths from the "naive" uncontrolled dynamics with $c=0$.
> We stress that the CDT static scheme aims to learn the initial value and control functions, which is different from a bootstrap particle filter (BPF) whose purpose is to perform nonlinear filtering.
> After training the initial value and control neural networks with the CDT static scheme, one can use them to approximate the FA-APF, and we refer to the resulting approximation of FA-APF as Static-APF. Our numerical experiments then compare Static-APF to the BPF as a benchmark. We have added a sentence to clarify this point.
> ***
> We hope we have managed to address your concerns about our work, and would be grateful if you could update your evaluation of our submission accordingly. Please feel free to leave us any additional comments or suggestions.
> ***
> [CMN2019] Chan-Wai-Nam Q, Mikael J, Warin X. Machine learning for semi linear PDEs. Journal of scientific computing. 2019 Jun;79(3):1667-712.
>
> [FPR2008] Fearnhead P, Papaspiliopoulos O, Roberts GO. Particle filters for partially observed diffusions. Journal of the Royal Statistical Society: Series B (Statistical Methodology). 2008 Sep;70(4):755-77.
>
> [FPRS2010] Fearnhead P, Papaspiliopoulos O, Roberts GO, Stuart A. Random‐weight particle filtering of continuous time processes. Journal of the Royal Statistical Society: Series B (Statistical Methodology). 2010 Sep;72(4):497-512.
>
> [SBBA2008] Snyder C, Bengtsson T, Bickel P, Anderson J. Obstacles to high-dimensional particle filtering. Monthly Weather Review. 2008 Dec;136(12):4629-40.
>
> [SBM2015] Snyder C, Bengtsson T, Morzfeld M. Performance bounds for particle filters using the optimal proposal. Monthly Weather Review. 2015 Nov;143(11):4750-61.

---

### Official Review · Reviewer_q4F1 · 2022-07-11

**Rating:** 6
**Confidence:** 4
**Soundness:** 3 good
**Presentation:** 2 fair
**Contribution:** 3 good

**Summary:**

The paper presents a novel method for online filtering where the latent process has a higher frequency than the observations. The authors define an algorithm, Computational Doob’s h-Transform, for iteratively learning a control function over the latent space and an observation to drive a stochastic process to a specific point at time T. The learned control function is then later applied in online filtering, where no further learning is required but the controlled dynamics can be applied to compute the particle filtering weights and to propagate the particles. CDT is applied to low-dimensional settings in the experiments section, where it is compared to bootstrap particle filtering for nonlinear dynamics, and to the exact analytical solution when available.

**Questions:**

Questions
1. On the iterative method: how many iterations of the iterative method were required until satisfactory results/convergence? Can you
      point to any theoretical results or give heuristics on the behaviour of the learned control through the iterative method?
2. On dimensionality: How does the method scale when the dimensionality of the problem is changed? I’m concerned of the learning of
       the control function, and how representative is the sampling scheme for observations during training when the problem is high-
      dimensional
3.  On experiments; Do they present a fair comparison? Is there some more recent work than bootstrap filters you could compare to in the
        experiment settings where no analytical solution is available?
4.  On training of neural networks; Did training the initial value and control networks simultaneously present any issues?
5.   On the loss function: the loss function considers only the final state of the value function. Was no regularization needed on the
         dynamics? Why, and are there some cases where it would be beneficial?
6. On the observations: There is a comment in the experiments section (Line 242) that the results are not sensitive to choice of T as long
        as the discretization has a small enough step size. What is small enough? A heuristic explanation/plot showing how long it takes the
       control to steer to the observation would have been informative.
7.   On clarity of text: (minor) some parts of the text had, to my knowledge, a number of typos and issues with grammar and notation. In
      addition, there was some vague phrasing. Please consider going through the following lines

Lines to check:
Line 13-14: applications listed, but no references provided
Line 21: “is collected” while referring to multiple observations
Line 29-30: “the goal of the article is…” This sentence felt too vague on my initial read
Line 33: “these quantities” which quantities are you referring to here? Only the control function has been referred to so far
Line 37: Efficient in what sense?
Line 44: “an homogenous” should be “a”?
Line 68: comma before while
Equation 4: why p_T? It is confusing to the reader in the case k> 1. The notation makes sense after reading 88-90, but this equation is before that
Line 136: “Ito lemma” should be “Ito’s lemma”
Line 143: Period missing from the last sentence of the paragraph.
Equation 15: t is used as time on the left-hand size, but it is the integrated variable on the right-hand side
Line 217: why is the control evaluated at t-t_k?
Line 238: odd to say that all experiments had 10 observations, when nearly all experiments vary K?
Line 248: why was it required to grow the number of samples when the number of observations grows? Does this mean that when K=1000, there were 100k samples (since there were 1k samples for K=10)
Line 255: “Analytically tractability”
Figure 1: the figure text was quite vague, I didn’t get what I should be concluding based on the plots? I can see that they somewhat match, but was there some additional insight I’m missing here?
Figure 2: The y-axis labels slightly overlap with the subplot borders


**Limitations:**

1. Societal impact of the work was not discussed. As the paper is presenting theoretical work, it does not require a deeper investigation into such matters.
2.  Scalability to higher dimensional problems and convergence of the iterative method were not sufficiently discussed or empirically tested, even though scalability to high-dimensional problems was explicitly stated as a motivating factor for not using standard numerical methods for solving PDEs (Line 126-128)


**Strengths And Weaknesses:**

Strengths
1. An original method for learning a control function for particle filtering using a Doob’s h-transform
2. Separation of learning the control and the filtering itself is an attractive idea for improving efficiency
3. The work is reproducible (the code was provided and I was able to run it)

Weaknesses
1. Limitations and related work were not discussed enough, the paper would have benefited from a well-structured Conclusions/Discussion
     section at the end instead of scattered remarks. It was not clear to me how the method presented compares to earlier work.
2. While I agree that covering the theory in detail is well justified, having three pages of background section would have been more
     comfortable to read if there was more text motivating why the topics covered are important or intuitive/heuristic explanations.
3.   The convergence properties of the iterative scheme were not studied empirically or theoretically, I was not able to develop an intuition
     into how the control behaves at iteration 1 versus iteration N.
4.   Low dimensional experiments, only comparisons to a simple baseline method (bootstrap particle filter) and not to more recent work
5.  The figures were focused on presenting error metrics, the behaviour of the learned control or the filtering distribution were hardly
      visualized

---

> ### Author Response · Authors · 2022-08-02
> **Response to Reviewer q4F1 [1 / 6]**
>
> We would like to thank the reviewer for the positive feedback of our work and the very thorough reading of our manuscript. His/her remarks and comments have helped us to significantly improve the quality of our manuscript. We give a point-by-point response below.
> ***
> 1. > Limitations and related work were not discussed enough, the paper would have benefited from a well-structured Conclusions/Discussion section at the end instead of scattered remarks. It was not clear to me how the method presented compares to earlier work.
>
> Thank you, this point was also raised by another reviewer and is now addressed in our revision. The main changes are as follows.
>
> A. A number of methodologies have been developed to remove (or asymptotically eliminate) the time-discretization bias; see e.g. [BPG2006, BPRF2006, BZ2020, FPR2008, FPRS2010]. To better position our work, we now made it clearer in the introduction and conclusion that this is not our
> objective: even if the control is learned perfectly, the SDE discretization bias remains. Although it is not always the case, in many situations, the time-discretization bias is negligible when compared to Monte Carlo
> errors. It is especially true in the regime of highly informative or extreme observations considered in this work.
>
> B. In the data-assimilation literature, a number of methods rely on Gaussianity assumptions (e.g. EnKF and related approaches such as [VL2010]). We now briefly describe some of these "asymptotically biased" methods, i.e. a possibly large, or difficult to quantify, bias can remain (in addition to the time-discretization bias) even as the number of particles goes to infinity. Although this class of methods is indeed extremely useful in very high-dimensional systems and spatially extended scenarios that are common in numerical weather forecasting, it is not the purpose of this article to tackle such problems. In contrast, the focus of this work is to offer a general methodology (by building on the strengths of neural networks) that can yield consistent estimates of the filtering distributions as both the number of discretization steps and the number of particles go to infinity. We have made these points clearer in our manuscript.
>
> C. We now discuss and numerically compare our proposed particle filters against the "Guided Intermediate Resampling Filters" (GIRF) of [DMM2015, PI2020]; see Point 4 below for our numerical findings. Like the bootstrap particle filter, these recently proposed algorithms move particles using the original SDE, but rely on resampling at intermediate time steps between observation times to guide particles to appropriate regions of the state-space. The choice of guiding functions that assign weights to particles at these intermediate steps is crucial for good algorithmic performance. We note that GIRF is in fact intimately related to Doob's $h$-transform as the optimal choice of guiding functions is given by the $h$-function [PI2020], i.e. $h(x,y,t)$ defined in Equation (10). However, even under this optimal choice, the resulting GIRF is still sub-optimal when compared to an auxiliary particle filter that move particles using the optimal control induced by the $h$-function, i.e. it is better to move particles well rather than rely on weighting and resampling. The latter behaviour is supported by our numerical experiments and a discussion of the above-mentioned connection has been added to the main text and appendix.
>
> D. We also have some discussion about future work in the newly-added conclusion.
> ***
> 2. > While I agree that covering the theory in detail is well justified, having three pages of background section would have been more comfortable to read if there was more text motivating why the topics covered are important or intuitive/heuristic explanations.
>
> We agree with this comment. Since models described by diffusion processes are very common in biological sciences (e.g. population modelling, multi-species models, stochastic delay population systems), neuroscience (e.g. models for synaptic input, stochastic Hodgkin–Huxley model, stochastic Fitzhugh–Nagumo model), and finance (e.g. modeling multi assets prices) - mainly because this flexible class of models is easily amenable to computations and simulations - we have motivated our study by taking several examples from these fields. In these disciplines, tracking signal from partial or noisy observations is a very common task.

---

> > ### Comment · Reviewer_q4F1 · 2022-08-08
> > **Response to rebuttal**
> >
> > I thank the authors for their thorough response. The rebuttals have included major improvements such as experiments in higher-dimensional settings, a more interesting baseline method than BPF and suggested content for a discussion section. As my main concerns on the manuscript have been properly addressed and questions extensively answered, I am changing my score to 6.

---

> > > ### Author Response · Authors · 2022-08-08
> > > **Reply to Reviewer q4F1**
> > >
> > > Thank you for taking the time to go through our rebuttal and for your kind response.

---

> ### Author Response · Authors · 2022-08-02
> **Response to Reviewer q4F1 [2 / 6]**
>
> We could have, indeed, also added a few examples from the geoscience literature. However, we feel that our proposed methodology (as it stands) does not scale up to the kind of dimensions that are practically relevant to geoscience (e.g. spatially extended fields with effective dimensionality >100 or >1000), unless we exploit certain model structure using techniques such as localization. Future work will explore how to tailor Doob's $h$-transform for problems in geoscience. As the camera-ready version allows an additional page of text, we will expand our introduction section to provide more motivation and references.
>
> To offer readers more intuition when going through the background section, we have added the following two figures (with links to our GitHub repository) to visualize the two main concepts underlying our work:
>
> A. [Doob's $h$-transform](https://anonymous.4open.science/r/CompDoobTransform/figures/illustrate_CDT.pdf);
>
> B. [using Doob's $h$-transform for online filtering of diffusions](https://anonymous.4open.science/r/CompDoobTransform/figures/illustrate_APF.pdf).
>
> Figure A shows how uncontrolled trajectories $X_t$ (*black*) under the original diffusion differs from controlled trajectories $\widehat{X}_t$ (*blue*) under the conditioned diffusion, induced by an informative observation $Y\sim\mathcal{N}(X_T,\sigma_Y^2)$ (*red*).
>
> For the purposes of online filtering, the bootstrap particle filter (BPF) and fully adapted auxiliary particle filter (FA-APF) require one to simulate trajectories under the original diffusion and conditioned diffusion respectively. Figure B illustrates the trajectories (*left panel*) and log-weights (*right panel*) generated by BPF (*black*) and FA-APF (*blue*). It is apparent that FA-APF allows one to obtain trajectories that are more consistent with informative data and weights that exhibit significantly less variability compared to the BPF.
>
> Our work proposes a new framework to compute Doob's $h$-transform that is tailored for the purpose of approximating the FA-APF in the setting of online filtering.
>
> Thank you again for raising this point that has helped us to improve our paper.
> ***
> 3. > The convergence properties of the iterative scheme were not studied empirically or theoretically, I was not able to develop an intuition into how the control behaves at iteration 1 versus iteration N.
>
> Thank you for this important comment. We have added numerical experiments to study convergence of the neural networks approximating the initial value and control functions as the training procedure progresses. These are compared to the optimal initial value function and optimal control function in the case of the Ornstein-Uhlenbeck model. The following figures (with links to our GitHub repository) illustrate quite rapid convergence. We also include the estimated loss function over the training iterations as these figures can be informative in practice. All figures and corresponding discussions about the training process have been added to both the main article and appendix.
>
> **Ornstein-Uhlenbeck model during initial training phase**
>
> OU-A. [Evolution of loss estimate over first $500$ optimization iterations](https://anonymous.4open.science/r/CompDoobTransform/figures/OU_loss_train.pdf)
>
> OU-B. [Evolution of neural network $\mathrm{N}_0(x,y)$ (*black to copper*) approximating the initial value function $v(x,y,0)$ (*red*) over first $500$ optimization iterations for a typical (*left panel*) and an extreme (*right panel*) observation $y$](https://anonymous.4open.science/r/CompDoobTransform/figures/OU_value_train.pdf)
>
> OU-C. [Evolution of neural network $-\mathrm{N}(x,y,t)$ (*black to copper*) approximating the control function $c_{\star}(x,y,t)$ (*red*) over first $500$ optimization iterations for a typical (*upper row*) and an extreme (*lower row*) observation $y$](https://anonymous.4open.science/r/CompDoobTransform/figures/OU_control_train.pdf)
>
> **Ornstein-Uhlenbeck model after training**
>
> OU-D. [Evolution of loss estimate over $2000$ optimization iterations under static (*red*) and iterative (*blue*) CDT schemes](https://anonymous.4open.science/r/CompDoobTransform/figures/OU_loss_compare.pdf)
>
> OU-E. [Neural network approximation $\mathrm{N}_0(x,y)$ of the initial value function $v(x,y,0)$ (*purple*) after training with the static (*red*) and iterative (*blue*) CDT schemes for a typical (*left panel*) and an extreme (*right panel*) observation $y$](https://anonymous.4open.science/r/CompDoobTransform/figures/OU_value_compare.pdf)
>
> OU-F. [Neural network approximation $-\mathrm{N}(x,y,t)$ of the control function $c_{\star}(x,y,t)$ (*purple*) after training with the static (*red*) and iterative (*blue*) CDT schemes for a typical (*upper row*) and an extreme (*lower row*) observation $y$](https://anonymous.4open.science/r/CompDoobTransform/figures/OU_control_compare.pdf)

---

> ### Author Response · Authors · 2022-08-02
> **Response to Reviewer q4F1 [3 / 6]**
>
> **Logistic diffusion model during initial training phase**
>
> LD-A. [Evolution of loss estimate over first $500$ optimization iterations](https://anonymous.4open.science/r/CompDoobTransform/figures/logistic_loss_train.pdf)
>
> LD-B. [Evolution of neural network $\mathrm{N}_0(x,y)$ (*black to copper*) approximating the initial value function $v(x,y,0)$ over first $500$ optimization iterations for a typical (*left panel*) and an extreme (*right panel*) observation $y$](https://anonymous.4open.science/r/CompDoobTransform/figures/logistic_value_train.pdf)
>
> LD-C. [Evolution of neural network $-\mathrm{N}(x,y,t)$ (*black to copper*) approximating the control function $c_{\star}(x,y,t)$ over first $500$ optimization iterations for a typical (*upper row*) and an extreme (*lower row*) observation $y$](https://anonymous.4open.science/r/CompDoobTransform/figures/logistic_control_train.pdf)
>
> **Logistic diffusion model after training**
>
> LD-D. [Evolution of loss estimate over $2000$ optimization iterations under static (*red*) and iterative (*blue*) CDT schemes and various levels of informative observations](https://anonymous.4open.science/r/CompDoobTransform/figures/logistic_loss.pdf)
>
> LD-E. [Neural network approximation $\mathrm{N}_0(x,y)$ of the initial value function $v(x,y,0)$ after training with the static (*red*) and iterative (*blue*) CDT schemes for a typical (*left panel*) and an extreme (*right panel*) observation $y$](https://anonymous.4open.science/r/CompDoobTransform/figures/logistic_value_compare.pdf)
>
> LD-F. [Neural network approximation $-\mathrm{N}(x,y,t)$ of the control function $c_{\star}(x,y,t)$ after training with the static (*red*) and iterative (*blue*) CDT schemes for a typical (*upper row*) and an extreme (*lower row*) observation $y$](https://anonymous.4open.science/r/CompDoobTransform/figures/logistic_control_compare.pdf)
>
> **Cell model after training**
>
> Cell-A. [Evolution of loss estimate over $2000$ optimization iterations under static (*red*) and iterative (*blue*) CDT schemes and various levels of informative observations](https://anonymous.4open.science/r/CompDoobTransform/figures/cell_loss.pdf)
>
> Cell-B. [Neural network approximation $\mathrm{N}_0(x,y)$ of the initial value function $v(x,y,0)$ after training with the static (*left column*) and iterative (*right column*) CDT schemes for a typical (*upper row*) and an extreme (*lower row*) observation $y$](https://anonymous.4open.science/r/CompDoobTransform/figures/cell_value_function.pdf)
>
> Cell-C. [Neural network approximation $-\mathrm{N}(x,y,t)$ of the control function $c_{\star}(x,y,t)$ after training with the static (*upper row*) and iterative (*lower row*) CDT schemes for a typical observation $y$](https://anonymous.4open.science/r/CompDoobTransform/figures/cell_control_function_in.pdf)
>
> Cell-D. [Neural network approximation $-\mathrm{N}(x,y,t)$ of the control function $c_{\star}(x,y,t)$ after training with the static (*upper row*) and iterative (*lower row*) CDT schemes for an extreme observation $y$](https://anonymous.4open.science/r/CompDoobTransform/figures/cell_control_function_out.pdf)
> ***
> 4. > Low dimensional experiments, only comparisons to a simple baseline method (bootstrap particle filter) and not to more recent work
>
> We thank the reviewer for this feedback that has helped us to significantly improve our numerical section.
>
> While applicability to higher-dimensional problems is indeed important, we would like to point out that online nonlinear filtering for diffusions is known to be already a hard problem in low dimensions. In low dimensions, even though one can run classical PDE solvers (e.g. finite difference and finite element methods), it is computationally impractical to use this for online filtering as one would require many solver calls (i.e. one call per observation). Hence we believe that our proposed approach based on a neural network solver is interesting even for low-dimensional problems. To illustrate the applicability of our approach to higher-dimensional settings where it is computationally infeasible to consider classical PDE solvers, we have added a numerical study of how our proposed and competing methods behave as the dimensionality increases.
>
> We have also added numerical comparisons to the recently proposed "Guided Intermediate Resampling Filters" (GIRF) of [DMM2015, PI2020], which are the current state-of-the-art within the class of asymptotically exact methods for filtering SDEs. The numerical results are given below (with links to our GitHub repository), and the corresponding findings have been added to our updated manuscript.

---

> ### Author Response · Authors · 2022-08-02
> **Response to Reviewer q4F1 [4 / 6]**
>
> **Ornstein-Uhlenbeck model**
>
> OU-A. [Results for Ornstein-Uhlenbeck model in dimension $d=1$ with various levels of informative observations](https://anonymous.4open.science/r/CompDoobTransform/figures/OU_results.pdf)
>
> OU-B. [Results for Ornstein-Uhlenbeck model with increasing dimension $d$ ](https://anonymous.4open.science/r/CompDoobTransform/figures/OU_dim_results.pdf)
>
> Here we have chosen to study the impact of increasing dimension on the Ornstein-Uhlenbeck model as analytical tractability in this example allows us to carefully benchmark our approximations of the fully adapted auxiliary particle filter (FA-APF). Moreover, this linear Gaussian setting is still challenging for many methods that do not exploit the model structure, and the ideal FA-APF itself suffers from the curse of dimensionality [SBBA2008, SBM2015] - as alluded to in Points 1 and 2.
>
> We have also leveraged analytical tractability in this model to implement the optimal guiding functions for GIRF, which is given by the Doob's $h$-function [PI2020]. These results show that GIRF can perform better than the bootstrap particle filter (BPF) in challenging regimes (i.e. when observations are informative or extreme, or in higher-dimensional settings), it is typically outperformed by the auxiliary particle filters.
>
> **Logistic diffusion model and cell model**
>
> LD-A. [Results for logistic diffusion model with various levels of informative observations](https://anonymous.4open.science/r/CompDoobTransform/figures/logistic_results.pdf)
>
> Cell-A. [Results for cell model with various levels of informative observations](https://anonymous.4open.science/r/CompDoobTransform/figures/cell_results.pdf)
>
> As the optimal guiding functions for GIRF is intractable for these two models, we considered a sub-optimal implementation that gradually introduces information from the next observation by annealing the observation density in a specific manner. We have added full implementation details in the appendix. In these numerical comparisons, we have considered both linear and quadratic annealing schedules which determine the rate at which information from the observation is introduced. Our numerical findings seem to indicate these instantiations of GIRF offer performance gains over the BPF when filtering extreme observations, but not when observations are informative. Moreover, Iterative-APF typically outperforms Static-APF and GIRF.
> ***
> 5. > The figures were focused on presenting error metrics, the behaviour of the learned control or the filtering distribution were hardly visualized.
>
> With reference to Point 3, we have added new figures to help readers visualize the neural networks approximating the initial value and control functions during the training procedure and after training is completed. These plots illustrate the rate of convergence during training and the differences between the neural networks obtained using the static and iterative CDT schemes.
>
> Is there any other set of experiments that the reviewer may find helpful to build more intuition?
>
> As for the filtering distributions, we did not find it very helpful to plot them but we could add some figures similar to the following illustrative plot if the reviewer thinks it would add value to the paper.
>
> [Using Doob's $h$-transform for online filtering of diffusions](https://anonymous.4open.science/r/CompDoobTransform/figures/illustrate_APF.pdf)
> ***
> 6. > On the iterative method: how many iterations of the iterative method were required until satisfactory results/convergence? Can you point to any theoretical results or give heuristics on the behaviour of the learned control through the iterative method?
>
> With reference to Point 3, we have added new figures showing:
>
> A. the behaviour of our neural network approximations during training (Figures OU-B, OU-C, LD-B, LD-C);
>
> B. the estimated loss function over the training iterations (Figures OU-D, LD-D, Cell-A).
>
> Although the methodology presented in our manuscript is quite close in spirit to several recently proposed neural network based backward SDE approaches to approximate semi-linear PDEs, most of these methods do not change the dynamics of the simulated sample paths as the training progresses. Hence most closely related theoretical results in the literature do not cover our approach.
> We are aware of only one relevant work [NR2021], which has some results concerning the stability of the training procedure near the optimal control and in high-dimensional settings. We will add this reference to our manuscript. Thank you for this question.
> ***

---

> ### Author Response · Authors · 2022-08-02
> **Response to Reviewer q4F1 [5 / 6]**
>
> 7. > How does the method scale when the dimensionality of the problem is changed? I’m concerned of the learning of the control function, and how representative is the sampling scheme for observations during training when the problem is high-dimensional
>
> We agree that scalability to higher-dimensional settings is indeed important.
> We have added a numerical study of how our proposed and competing methods behave as the dimensionality increases, and refer to Point 4 for more details.
> ***
> 8. > On experiments; Do they present a fair comparison? Is there some more recent work than bootstrap filters you could compare to in the experiment settings where no analytical solution is available?
>
> To the best of our knowledge, there are relatively few existing methodologies that are applicable in the setup we consider and are as general (i.e. do not heavily rely on the structure of the model) as ours. Note that our method does not exploit any structure of the diffusion except the ability to simulate SDEs and differentiate through the transition densities.
>
> With reference to Point 4, we have added numerical comparisons to the recently proposed "Guided Intermediate Resampling Filters" (GIRF) of [DMM2015, PI2020], which are the current state-of-the-art within the class of asymptotically exact methods for filtering SDEs. Although GIRF is generally applicable, the choice of guiding functions that assign weights to particles at intermediate steps is crucial for good algorithmic performance. Our implementation of GIRF employed the optimal guiding functions for the Ornstein-Uhlenbeck model, and a sub-optimal but practical choice for other models that gradually introduces information from the next observation by annealing the observation density in a specific manner. We have also considered various annealing schedules (but report just two) in an effort to provide a fair comparison.
>
> If the reviewer is aware of another class of methods that is applicable to this problem setting, we would be happy to perform additional comparisons.
> ***
> 9. > On training of neural networks; Did training the initial value and control networks simultaneously present any issues?
>
> We did not encounter any numerical instabilities in our experiments. Interestingly, recent work by [NR2021] (from the backward SDE and PDE perspective), indicate that faster convergence of the training procedure could be achieved by using different parameterizations of the neural networks approximating the initial value and control functions. We will experiment with these alternative formulations in future work.
> ***
> 10. > On the loss function: the loss function considers only the final state of the value function. Was no regularization needed on the dynamics? Why, and are there some cases where it would be beneficial?
>
> It is an important point which we have also given much thought to when designing the method. We were initially surprised that there is enough "gradient information" for the optimal control to be learned efficiently even though the "loss function only depends on the final state of the value function". Empirically, this does not seem to be an issue.
>
> One instructive way to understand why is to view the CDT algorithm as minimizing some divergence between path measures defined on path space [NR2021]. We refer also to the excellent article by [CWN2019] for discussions and comparisons of neural architecture and loss functions (in the context of solving semi-linear PDEs).
> ***
> 11. > On the observations: There is a comment in the experiments section (Line 242) that the results are not sensitive to choice of T as long as the discretization has a small enough step size. What is small enough? A heuristic explanation/plot showing how long it takes the control to steer to the observation would have been informative.
>
> Thank you for this comment. When the time-discretization is too coarse, the induced bias will dominate the total error (and hence poor performance is obtained) and the time-discretized process might be unstable to simulate. What is considered "small enough" depends on the particular model and the informativeness of the observations. We advocate running the methodology for increasingly finer discretization: all the performance metrics quickly stabilize when the discretization becomes "small enough". We have not been able to find a more theoretically grounded approach for setting the step-size, and have been quite satisfied with this data-driven approach. We have added a sentence in the main text to better explain this point and our rationale. Lastly, the requested plots have been added - we refer to Figures A and B in Point 2.
> ***

---

> ### Author Response · Authors · 2022-08-02
> **Response to Reviewer q4F1 [6 / 6]**
>
> 12. > On clarity of text: (minor) some parts of the text had, to my knowledge, a number of typos and issues with grammar and notation. In addition, there was some vague phrasing. Please consider going through the following lines
>
> Thank you for your very careful review of our manuscript, it is very much appreciated. We have corrected all the points you have raised - this has significantly improved the quality of the text. Thank you!
> ***
> We hope we have managed to address your concerns about our work, and would be grateful if you could update your evaluation of our submission accordingly. Please feel free to leave us any additional comments or suggestions.
> ***
> [BPG2006] Beskos A, Papaspiliopoulos O, Roberts GO. Retrospective exact simulation of diffusion sample paths with applications. Bernoulli. 2006 Dec;12(6):1077-98.
>
> [BPRF2006] Beskos A, Papaspiliopoulos O, Roberts GO, Fearnhead P. Exact and computationally efficient likelihood‐based estimation for discretely observed diffusion processes (with discussion). Journal of the Royal Statistical Society: Series B (Statistical Methodology). 2006 Jun;68(3):333-82.
>
> [BZ2020] Blanchet J, Zhang F. Exact simulation for multivariate Itô diffusions. Advances in Applied Probability. 2020 Dec;52(4):1003-34.
>
> [CWN2019] Chan-Wai-Nam Q, Mikael J, Warin X. Machine learning for semi linear PDEs. Journal of scientific computing. 2019 Jun;79(3):1667-712.
>
> [DMM2015] Del Moral P, Murray LM. Sequential Monte Carlo with highly informative observations. SIAM/ASA Journal on Uncertainty Quantification. 2015;3(1):969-97.
>
> [FPR2008] Fearnhead P, Papaspiliopoulos O, Roberts GO. Particle filters for partially observed diffusions. Journal of the Royal Statistical Society: Series B (Statistical Methodology). 2008 Sep;70(4):755-77.
>
> [FPRS2010] Fearnhead P, Papaspiliopoulos O, Roberts GO, Stuart A. Random‐weight particle filtering of continuous time processes. Journal of the Royal Statistical Society: Series B (Statistical Methodology). 2010 Sep;72(4):497-512.
>
> [NR2021] Nüsken N, Richter L. Solving high-dimensional Hamilton–Jacobi–Bellman PDEs using neural networks: perspectives from the theory of controlled diffusions and measures on path space. Partial Differential Equations and Applications. 2021 Aug;2(4):1-48.
>
> [PI2020] Park J, Ionides EL. Inference on high-dimensional implicit dynamic models using a guided intermediate resampling filter. Statistics and Computing. 2020 Sep;30(5):1497-522.
>
> [SBBA2008] Snyder C, Bengtsson T, Bickel P, Anderson J. Obstacles to high-dimensional particle filtering. Monthly Weather Review. 2008 Dec;136(12):4629-40.
>
> [SBM2015] Snyder C, Bengtsson T, Morzfeld M. Performance bounds for particle filters using the optimal proposal. Monthly Weather Review. 2015 Nov;143(11):4750-61.
>
> [VL2010] van Leeuwen, P.J. (2010) Nonlinear data assimilation in geosciences: an extremely efficient particle filter. Quarterly Journal of the Royal Meteorological Society, 136, 1991–1999.

---

### Official Review · Reviewer_mpC8 · 2022-07-12

**Rating:** 6
**Confidence:** 4
**Soundness:** 3 good
**Presentation:** 4 excellent
**Contribution:** 3 good

**Summary:**

The paper addresses the task of online filtering for discretely observed nonlinear diffusion approaches. The paper proposes a method based on the fully adapted auxiliary particle filter (FA-APF). The paper discusses how a control function can be used to construct a controlled diffusion that takes into account a future observation. The desired control function is related to Doob’s h-transform and can be obtained by solving the backward Kolmogorov equations. This is computationally intractable, so the proposed strategy is to work with the value function (the negative log of h). The optimal control can then be expressed in terms of the value function and approximated with neural networks. These networks can be trained by simulating the controlled diffusion trajectories. After training, the online filtering can be performed using the approximately optimal control.

The paper presents numerical results for three models: an Ornstein-Uhlenbeck model, a logistic diffusion model, and a cell model. The results are compared to a bootstrap particle filter. For the Ornstein-Uhlenbeck model, which is analytically tractable, there is also comparison with the exact APF. There superiority of the proposed method is clear in all three examples.


**Questions:**

1.	How many particles were used in the experiments (or what was the scaling factor with K)? Why was this value chosen? What is the sensitivity to this choice? Is it possible to achieve improved performance with more particles or is the computational burden already considerable, i.e. if I were trying to implement a real-time system with a reasonably powerful computer, with a reasonable time between observations, how many particles could I use?

2.	What is the computational overhead? How does the computation of the proposed method compare to the BPF? What are the key contributors to the computational burden?

3.	What are the state-of-the-art competing techniques? Why are these not considered in a quantitative performance comparison? Are methods like [R1] inapplicable for the studied settings?


**Limitations:**

Yes.

**Strengths And Weaknesses:**

Strengths
1.	The paper addresses the important problem of online inference for diffusion processes (although it doesn’t make a compelling case as to why this problem is important) and provides an elegant algorithm for performing the filtering. To the best of my knowledge, this represents a novel approach to solving this problem and I would rate it as a highly original approach. I think the work could have a significant impact on research addressing online inference for diffusion processes.

2.	The paper is very well written. The material is relatively dense, but the authors do an excellent job of providing a clear description of the algorithm. It is developed step-by-step in an intuitive manner.


Weaknesses
1.	Related work and performance comparison: there is very little discussion of existing methods for filtering partially observed diffusions. Multiple methods are acknowledged (and dismissed) with the single sentence “Specialized methodologies have been developed to circumvent or mitigate these issues.” Other methods like [R1] are ignored.
Even if some of the methods are more limited in their applicability, there should have been an effort to compare with them experimentally for examples where they are applicable. Some discussion of the limitations of existing work should be included in the paper.

2.	Motivation: The paper does not provide a compelling motivation for the analyzed problem. There are very general claims about the ubiquity of diffusion processes in the first few sentences but no citations to any literature to support these claims. There are no citations of work providing more practical examples of where the proposed filtering problem would prove beneficial. The studied examples, while providing a good illustration of the benefits of the technique, do not provide compelling evidence of the practical utility. The cell model and the logistic diffusion model for population size dynamics do not seem suited to a meaningful, practical filtering task.

3.	Experimental analysis: There is no performance comparison to any methods except for the basic bootstrap particle filter. There have been multiple approaches to address the filtering task, so there should either be a clear explanation as to why existing approaches are inapplicable or other methods should be included. I could not find the details of exactly how many particles are used in the simulations (apart from the specification that it increases linearly with K). This seems to be an important design choice and there should be an explanation as to how the value was selected. Preferably there should be an analysis of how performance changes with varying K. There is no reporting of computational requirements. It is not clear if the proposed filter and the BPF have the same (or very similar) computations. Presumably there are additional neural network evaluations for the proposed scheme, but these may be negligible compared to other computational overhead. If they are non-negligible, then a fairer comparison would use more particles for the BPF.

[R1] Jasra, A., Law, K. J., & Yu, F. (2020). Unbiased filtering of a class of partially observed diffusions. Advances in Applied Probability, 1-27.

---

> ### Author Response · Authors · 2022-08-02
> **Response to Reviewer mpC8 [1 / 4]**
>
> We would like to thank the reviewer for the positive feedback of our work and the very thorough reading of our manuscript. His/her remarks and comments have helped us to significantly improve the quality of our manuscript. We give a point-by-point response below.
> ***
> 1. > Weaknesses 1. Related work and performance comparison: there is very little discussion of existing methods for filtering partially observed diffusions. Multiple methods are acknowledged (and dismissed) with the single sentence “Specialized methodologies have been developed to circumvent or mitigate these issues.” Other methods like [R1] are ignored. Even if some of the methods are more limited in their applicability, there should have been an effort to compare with them experimentally for examples where they are applicable. Some discussion of the limitations of existing work should be included in the paper.
>
> Thank you for this comment. It is a very important point that was also raised by another reviewer.
>
> A. A number of methodologies have been developed to remove (or asymptotically eliminate) the time-discretization bias; see e.g. [BPG2006, BPRF2006, BZ2020, FPR2008, FPRS2010] that were cited in our manuscript.
> To better position our work, we now made it clearer in the introduction that this is not our objective: even if the control is learned perfectly, the SDE discretization bias remains. Although it is not always the case, in many situations, the time-discretization bias is negligible when compared to Monte Carlo errors. It is especially true in the regime of highly informative or extreme observations considered in this work.
>
> We thank the reviewer for pointing us to the reference [JLY2020], which is very relevant to the setup we have considered and complementary to our proposed method. In particular, one can think of the novel double randomization scheme of [JLY2020] as a "wrapper" to enable our proposed auxiliary particle filters to return estimators of filtering expectations with no time-discretization bias. We now cite this reference and discuss this connection.
>
> B. When assimilating high-dimensional systems, there are indeed a number of methods relying on Gaussianity assumptions (e.g. EnKF and related approaches). We now briefly describe some of these methods and also emphasize that we do not compare to these methods that are "asymptotically biased", i.e. a possibly large, or difficult to quantify, bias can remain (in addition to the time-discretization bias) even as the number of particles goes to infinity. In contrast, the focus of this work is to offer a general methodology (by building on the strengths of neural networks) that can yield consistent estimates of the filtering distributions as both the number of discretization steps and the number of particles go to infinity.
>
> C. We have now added a brief discussion of the method pioneered by [VL2010] that attempts to steer particles toward the observations and make their weights exhibit less variance. This method relies on some model structure, e.g. the observation is assumed to be of the form $y = F(x) + \varepsilon$ where $\varepsilon$ denotes Gaussian noise, and has a number of tuning parameters.  Furthermore, it can be understood as parametrizing a linear control, which is only expected to work well for linear Gaussian dynamics (which is a very important special case in applications). Indeed, this method appears to work very well in challenging geoscience applications. It is now discussed within the main text.
>
> D. We also now briefly discuss the "Implicit Particle Filter" of [CMT2010] in our updated manuscript. The method attempts to transform standard Gaussian samples (by solving a nonlinear optimization routine) to create samples from the optimal proposal density. Operationalizing this methodology requires a number of assumptions and requires a nonlinear optimization step for each particle and each time step, which can quickly become computational burdensome.

---

> > ### Comment · Reviewer_mpC8 · 2022-08-08
> > **Response to rebuttal**
> >
> > The response to the reviews has been very thorough. The proposed changes lead to a considerably stronger paper with a much better performance evaluation. In my view, most of the concerns raised by other reviewers have been addressed, leading to improvements and better explanations concerning the proposed method. One consideration is that the changes from the original submission are substantial and the revised paper almost needs to go through another thorough review. My questions have been satisfactorily addressed and I have raised my score to a 6.

---

> > > ### Author Response · Authors · 2022-08-09
> > > **Reply to Reviewer mpC8**
> > >
> > > Thank you for taking the time to go through our rebuttal and for your kind response.

---

> ### Author Response · Authors · 2022-08-02
> **Response to Reviewer mpC8 [2 / 4]**
>
> E. We now discuss and numerically compare our proposed particle filters against the "Guided Intermediate Resampling Filters" (GIRF) of [DMM2015, PI2020]; see Point 3 below for our numerical findings. Like the bootstrap particle filter, these recently proposed algorithms move particles using the original SDE, but rely on resampling at intermediate time steps between observation times to guide particles to appropriate regions of the state-space. The choice of guiding functions that assign weights to particles at these intermediate steps is crucial for good algorithmic performance. We note that GIRF is in fact intimately related to Doob's $h$-transform as the optimal choice of guiding functions is given by the $h$-function [PI2020], i.e. $h(x,y,t)$ defined in Equation (10). However, even under this optimal choice, the resulting GIRF is still sub-optimal when compared to an auxiliary particle filter that move particles using the optimal control induced by the $h$-function, i.e. it is better to move particles well rather than rely on weighting and resampling. The latter behaviour is supported by our numerical experiments and a discussion of the above-mentioned connection has been added to the main text and appendix.
> ***
> 2. > Motivation: The paper does not provide a compelling motivation for the analyzed problem. There are very general claims about the ubiquity of diffusion processes in the first few sentences but no citations to any literature to support these claims. There are no citations of work providing more practical examples of where the proposed filtering problem would prove beneficial. The studied examples, while providing a good illustration of the benefits of the technique, do not provide compelling evidence of the practical utility. The cell model and the logistic diffusion model for population size dynamics do not seem suited to a meaningful, practical filtering task.
>
> We agree with this comment. Since models described by diffusion processes are very common in biological sciences (e.g. population modelling, multi-species models, stochastic delay population systems), neuroscience (e.g. models for synaptic input, stochastic Hodgkin–Huxley model, stochastic Fitzhugh–Nagumo model), and finance (e.g. modeling multi assets prices) - mainly because this flexible class of models is easily amenable to computations and simulations - we have motivated our study by taking several examples from these fields.
> In these disciplines, tracking signal from partial or noisy observations is a very common task.
>
> We could have, indeed, also added a few examples from the geoscience literature. However, we feel that our proposed methodology (as it stands) does not scale up to the kind of dimensions that are practically relevant to geoscience
> (e.g. spatially extended fields with effective dimensionality >100 or >1000), unless we exploit certain model structure using techniques such as localization. Future work will explore how to tailor Doob's $h$-transform for problems in geoscience.
>
> As the camera-ready version allows an additional page of text, we will expand our introduction section to provide more motivation and references. Thank you again for raising this point.
> ***
> 3. >There is no performance comparison to any methods except for the basic bootstrap particle filter.
>
> We thank the reviewer for this comment that has helped us to significantly improve our numerical section. We have added numerical comparisons to the recently proposed "Guided Intermediate Resampling Filters" (GIRF) of [DMM2015, PI2020], which are the current state-of-the-art within the class of asymptotically exact methods for filtering SDEs. The numerical results are given below (with links to our GitHub repository), and the corresponding findings have been added to our updated manuscript.

---

> ### Author Response · Authors · 2022-08-02
> **Response to Reviewer mpC8 [3 / 4]**
>
> **Ornstein-Uhlenbeck model**
>
> OU-A. [Results for Ornstein-Uhlenbeck model in dimension $d=1$ with various levels of informative observations](https://anonymous.4open.science/r/CompDoobTransform/figures/OU_results.pdf)
>
> OU-B. [Results for Ornstein-Uhlenbeck model with increasing dimension $d$](https://anonymous.4open.science/r/CompDoobTransform/figures/OU_dim_results.pdf)
>
> Following the feedback from other reviewers, we have added a numerical study of how our proposed and competing methods behave as the dimensionality increases. Here we have chosen to perform this study on the Ornstein-Uhlenbeck model as analytical tractability in this example allows us to carefully benchmark our approximations of the fully adapted auxiliary particle filter (FA-APF). Moreover, this linear Gaussian setting is still challenging for many methods that do not exploit the model structure, and the ideal FA-APF itself suffers from the curse of dimensionality [SBBA2008, SBM2015] - as alluded to in Point 2.
>
> We have also leveraged analytical tractability in this model to implement the optimal guiding functions for GIRF, which is given by the Doob's $h$-function [PI2020]. These results show that GIRF can perform better than the bootstrap particle filter (BPF) in challenging regimes (i.e. when observations are informative or extreme, or in higher-dimensional settings), and it is typically outperformed by the auxiliary particle filters.
>
> **Logistic diffusion model and cell model**
>
> LD-A. [Results for logistic diffusion model with various levels of informative observations](https://anonymous.4open.science/r/CompDoobTransform/figures/logistic_results.pdf)
>
> Cell-A. [Results for cell model with various levels of informative observations](https://anonymous.4open.science/r/CompDoobTransform/figures/cell_results.pdf)
>
> As the optimal guiding functions for GIRF is intractable for these two models, we considered a sub-optimal implementation that gradually introduces information from the next observation by annealing the observation density in a specific manner. We have added full implementation details in the appendix. In these numerical comparisons, we have considered both linear and quadratic annealing schedules which determine the rate at which information from the observation is introduced. Our numerical findings seem to indicate these instantiations of GIRF offer performance gains over the BPF when filtering extreme observations, but not when observations are informative. Moreover, Iterative-APF typically outperforms Static-APF and GIRF.
> ***
> 4. > I could not find the details of exactly how many particles are used in the simulations (apart from the specification that it increases linearly with K). This seems to be an important design choice and there should be an explanation as to how the value was selected. Preferably there should be an analysis of how performance changes with varying K.
> How many particles were used in the experiments (or what was the scaling factor with K)? Why was this value chosen? What is the sensitivity to this choice? Is it possible to achieve improved performance with more particles or is the computational burden already considerable, i.e. if I were trying to implement a real-time system with a reasonably powerful computer, with a reasonable time between observations, how many particles could I use?
>
> In all numerical experiments, we chose the number of particles $N$ as $N=2^6K/100$, where $K$ denotes the number of observations. Increasing $N$ linearly with $K$ is necessary to keep marginal likelihood estimators stable [BDMD2014] when producing the figures listed in Point 3. Our numerical findings are not sensitive to the choice of scaling factor $2^6/100$ (as long as this is large enough to have adequate number of particles for the smallest $K$ considered); it was simply chosen for ease of plotting the results, as too large a factor will render it hard to visualize the differences between the particle filters (in terms of the ELBO gap and the variance of the log-marginal likelihood estimators). For a given model and application, the appropriate scaling factor, or equivalently the number of particles required, would be very much problem-specific, and dependent on the desired precision of filtering approximations. Hence the number of particles is usually determined in practice by preliminary runs. Furthermore, as pointed out by the reviewer, how the computational cost depends on the number of particles also depends on the specificities of the problem. For the models we have considered, it was reasonable to use quite large number of particles compared to what is usually possible in geoscience applications (as discussed in Point 2).
> ***

---

> ### Author Response · Authors · 2022-08-02
> **Response to Reviewer mpC8 [4 / 4]**
>
> 5. > There is no reporting of computational requirements. It is not clear if the proposed filter and the BPF have the same (or very similar) computations. Presumably there are additional neural network evaluations for the proposed scheme, but these may be negligible compared to other computational overhead. If they are non-negligible, then a fairer comparison would use more particles for the BPF.
> What is the computational overhead? How does the computation of the proposed method compare to the BPF? What are the key contributors to the computational burden?
>
> The main computational overhead comes from training the initial value and control neural networks; the additional overhead from having to evaluate these neural networks in our proposed particle filters was negligible. Following this comment, we now detail the computational cost for each model in the appendix. The main point here is that the training cost is actually very small when filtering long time series, or running BPF with many particles in challenging regimes.
> ***
> We hope we have managed to address your concerns about our work, and would be grateful if you could update your evaluation of our submission accordingly. Please feel free to leave us any additional comments or suggestions.
> ***
> [BDMD2014] Bérard J, Del Moral P, Doucet A. A lognormal central limit theorem for particle approximations of normalizing constants. Electronic Journal of Probability. 2014 Jan;19:1-28.
>
> [BPG2006] Beskos A, Papaspiliopoulos O, Roberts GO. Retrospective exact simulation of diffusion sample paths with applications. Bernoulli. 2006 Dec;12(6):1077-98.
>
> [BPRF2006] Beskos A, Papaspiliopoulos O, Roberts GO, Fearnhead P. Exact and computationally efficient likelihood‐based estimation for discretely observed diffusion processes (with discussion). Journal of the Royal Statistical Society: Series B (Statistical Methodology). 2006 Jun;68(3):333-82.
>
> [BZ2020] Blanchet J, Zhang F. Exact simulation for multivariate Itô diffusions. Advances in Applied Probability. 2020 Dec;52(4):1003-34.
>
> [CMT2010] Chorin, A.J., Morzfeld, M. and Tu, X. (2010) Interpolation and iteration for nonlinear filters. Communications in Applied Mathematics and Computational Science, 5, 221–240.
>
> [DMM2015] Del Moral P, Murray LM. Sequential Monte Carlo with highly informative observations. SIAM/ASA Journal on Uncertainty Quantification. 2015;3(1):969-97.
>
> [FPR2008] Fearnhead P, Papaspiliopoulos O, Roberts GO. Particle filters for partially observed diffusions. Journal of the Royal Statistical Society: Series B (Statistical Methodology). 2008 Sep;70(4):755-77.
>
> [FPRS2010] Fearnhead P, Papaspiliopoulos O, Roberts GO, Stuart A. Random‐weight particle filtering of continuous time processes. Journal of the Royal Statistical Society: Series B (Statistical Methodology). 2010 Sep;72(4):497-512.
>
> [JLY2020] Jasra A, Law KJ, Yu F. Unbiased filtering of a class of partially observed diffusions. Advances in Applied Probability. 2020:1-27.
>
> [PI2020] Park J, Ionides EL. Inference on high-dimensional implicit dynamic models using a guided intermediate resampling filter. Statistics and Computing. 2020 Sep;30(5):1497-522.
>
> [SBBA2008] Snyder C, Bengtsson T, Bickel P, Anderson J. Obstacles to high-dimensional particle filtering. Monthly Weather Review. 2008 Dec;136(12):4629-40.
>
> [SBM2015] Snyder C, Bengtsson T, Morzfeld M. Performance bounds for particle filters using the optimal proposal. Monthly Weather Review. 2015 Nov;143(11):4750-61.
>
> [VL2010] van Leeuwen, P.J. (2010) Nonlinear data assimilation in geosciences: an extremely efficient particle filter. Quarterly Journal of the Royal Meteorological Society, 136, 1991–1999.

---

### Official Review · Reviewer_qRy1 · 2022-07-15

**Rating:** 6
**Confidence:** 2
**Soundness:** 3 good
**Presentation:** 2 fair
**Contribution:** 3 good

**Summary:**

Disclaimer: I am not expert in SDE theory so my review will focus on the high level and clarity.

The authors are interested in the filtering problem in SDEs, i.e. computing the conditional distributions x_t|y<t, with x the latent process and y the observations.
The focus is on particule filtering, where the filtering distribution is approximated via discrete particules.
In particule filtering, a guiding function q or proposal is used to propagate the values of the particules in time.
A convenient guide is the 'Fully Adapted Auxiliary Particle Filter' which samples from the conditional p(x_t+1|y_t+1, x_t).

Doob's theorem happens to give an expression for the SDE corresponding to this conditional p(x_t+1|y_t+1, x_t), which shares the same
diffusion as the a priori SDE and whose drift function is equal to the prior drift function plus an offest, or control function.
Sampling from this SDE for starting point x_t would give the desired sample of p(x_t+1|y_t+1, x_t) for the purpose of particule filtering

The authors propose an algorithm to find learn this control function c, without having to have to first solve a PDE first.
This function and the prior density is parameterized by neural network.
An objective is proposed to learn the control function.
An algorithm is proposed to optimize this objective.

The algorithm is evaluated to check the validity of the learned control in toy settings and in Particle filtering scenarios.


**Questions:**

Some comments / questions:

two related comments first
* eta_x should depend on the conditioning end observation.
If you drop this dependency, you are not sampling from the conditioned SDE.
In your experiment (L252), you drop this dependency. Is this a typo, a design choice, an approximation?

* L159: "In practice, we will let the three sources of randomness be independent of each other."
What is lost with that approximation (is it an approximation)? If you choose T large, then the process reverts to the prior (independent of Y), stationary distribution of the uncontrolled SDE.
Is it what you do?

* The result section, shows that the learned control matches the exact control on a linear SDE scenario.
Many details are missing, an appendix with those would be great.
as a reader, I don't want to have to fill all the gaps. Even less as a reviewer.
For example, for OU, you could give the exact expression for h, and v.

* The proposed objective seems to contain the control on both side of the square loss, the V_T and the log g(..).
I do not really understand or see why the optimal control is a minimizer of objective.
It also feels like hitting a moving target.

* It would be good to have a discussion of this optimization problem.
For example, how do the dynamics of learning look like?

* Could you be more precise on which variables you are taking expectations over and why?
This is important for the loss.

* Also could you give a justification for the 'stop gradient' comment?
You have a loss that has multiple dependence on c and you decide to only propagate through one part.

* CDT algorithm, it would help to have a more detailed pseudo code for the algorithm.

* Doob's transform: I suggest you either give more details or remove the sketch from the main paper. As is it is not useful.

* Where is the conclusion? It is good to end a paper with the conclusion summarizing the method and results.



**Strengths And Weaknesses:**

I find the proposed manuscript is well motivated.
The background on particule filtering is clear.

My SDE theory is less developed but I don't understand section 2.5,
not even at a high level. As a result I m not sure I trust the content of section 3.1 (the loss and algorithm)
I m possibly not the best reviewer (hence the low confidence)
but I also think clarity could be greatly improved (see below).

---

> ### Author Response · Authors · 2022-08-02
> **Response to Reviewer qRy1 [1 / 4]**
>
> We would like to thank the reviewer for his/her comments that helped us improve the clarity of our manuscript and make it more accessible to readers less versed in SDE theory. The comments are well received and were very helpful. We give a point-by-point response to these comments below.
> ***
> 1. > My SDE theory is less developed but I don't understand section 2.5, not even at a high level. As a result I m not sure I trust the content of section 3.1 (the loss and algorithm) I m possibly not the best reviewer (hence the low confidence) but I also think clarity could be greatly improved (see below).
>
> To help readers who are less familiar with SDEs, we have added the following two figures (with links to our GitHub repository) to visualize the two main concepts underlying our work:
>
> A. [Doob's $h$-transform](https://anonymous.4open.science/r/CompDoobTransform/figures/illustrate_CDT.pdf);
>
> B. [using Doob's $h$-transform for online filtering of diffusions](https://anonymous.4open.science/r/CompDoobTransform/figures/illustrate_APF.pdf).
>
> Figure A shows how uncontrolled trajectories $X_t$ (*black*) under the original diffusion differs from controlled trajectories $\widehat{X}_t$ (*blue*) under the conditioned diffusion, induced by an informative observation $Y\sim\mathcal{N}(X_T,\sigma_Y^2)$ (*red*).
>
> For the purposes of online filtering, the bootstrap particle filter (BPF) and fully adapted auxiliary particle filter (FA-APF) require one to simulate trajectories under the original diffusion and conditioned diffusion respectively. Figure B illustrates the trajectories (*left panel*) and log-weights (*right panel*) generated by BPF (*black*) and FA-APF (*blue*). It is apparent that FA-APF allows one to obtain trajectories that are more consistent with informative data and weights that exhibit significantly less variability compared to the BPF.
>
> Our work proposes a new framework to compute Doob's $h$-transform that is tailored for the purpose of approximating the FA-APF in the setting of online filtering.
> ***
> 2. > $\eta_x$ should depend on the conditioning end observation. If you drop this dependency, you are not sampling from the conditioned SDE. In your experiment (L252), you drop this dependency. Is this a typo, a design choice, an approximation?
>
> We agree that, if our goal were to sample a diffusion process starting from some known distribution $\pi_0$ at time $t=0$, conditioned upon an observation $y_T$ collected at time $T$, the law of the conditional diffusion at time $t=0$ would depend upon the observation $y_T$. In the "online filtering" context of our manuscript, and as rightfully described by the reviewer, we are interested in describing the dynamics of the conditioned diffusion: the initial (conditioned) distribution is not really an object of interest (as can be seen from the description of the fully adapted auxiliary particle filter"). The distribution denoted as $\eta_x$ is only used as a "computational device" during the training phase of the methodology and does not need to be equal to the actual conditioned distribution. As a matter of fact and as empirically observed, for computational stability, the distribution $\eta_x$ should be chosen to be more "spread out" than the actual conditioned distribution. We have now made this point clearer in the updated manuscript.
> ***
> 3. > L159: "In practice, we will let the three sources of randomness be independent of each other." What is lost with that approximation (is it an approximation)?
>
> It is related to the previous Point 2: these distributions do not need to be equal to the actual conditioned distribution. The randomness is used for exploring the state-space and learning the control in the training phase of the proposed methodology. Our proposed approach of choosing the three sources of randomness is easy to implement and works well empirically. Importantly, this does *not* lead to a biased or or an uncontrolled approximation: we have now made this important point clearer.
> ***
> 4. > The result section, shows that the learned control matches the exact control on a linear SDE scenario. Many details are missing, an appendix with those would be great. as a reader, I don't want to have to fill all the gaps. Even less as a reviewer. For example, for OU, you could give the exact expression for $h$, and $v$.
>
> Thank you for this suggestion. We have now added some details in the main text and provide the details of the Ornstein-Uhlenbeck model in the appendix.
> ***

---

> ### Author Response · Authors · 2022-08-02
> **Response to Reviewer qRy1 [2 / 4]**
>
> 5. > The proposed objective seems to contain the control on both side of the square loss, the $V_T$ and the $\log g(..)$. It also feels like hitting a moving target.
>
> Thank you for this comment. We have now made this crucial point much clearer. In Equation (18), the control $c$ is to be considered *fixed* (as zero in the "CDT static scheme" or as a current approximation of the control in the "CDT iterative scheme"). Only the weights of the neural networks $\textrm{N}_0$ and $\textrm{N}$ in Equation (16) are being learned. Consequently, the dynamics of $X^{c,y}_t$ is considered fixed, and only the dynamics of the process $V_t$ depends on the neural networks $\textrm{N}_0$ and $\textrm{N}$. In summary, in the difference $V_T + \log g(X^{c,Y}_T, Y)$, only the term $V_T$ is "moving".
> ***
> 6. > Also could you give a justification for the ’stop gradient’ comment?
>
> It is related to the previous point 5. In practice, we advocate using the "CDT iterative scheme" which chooses the control $c$ at each iteration of our training procedure as the current neural network approximation of the control, i.e. setting $c(x,y,t)=-\textrm{N}(x,y,t)$.
> As explained in the previous comment, the control $c$ is assumed to be fixed when constructing the loss in Equation (18).
> Consequently, it is necessary to use a "stop-gradient" operation to ensure that gradient operations are not propagated through the control function $c$. Points 5 and 6 are important and we thank the reviewer for raising these questions. We have now made the manuscript much clearer on these points.
> ***
> 7. > It would be good to have a discussion of this optimization problem. For example, how do the dynamics of learning look like?
>
> Thank you for this suggestion. We have added the following figures (with links to our GitHub repository) and corresponding discussions about the training process.
>
> **Ornstein-Uhlenbeck model during initial training phase**
>
> OU-A. [Evolution of loss estimate over first $500$ optimization iterations](https://anonymous.4open.science/r/CompDoobTransform/figures/OU_loss_train.pdf)
>
> OU-B. [Evolution of neural network $\mathrm{N}_0(x,y)$ (*black to copper*) approximating the initial value function $v(x,y,0)$ (*red*) over first $500$ optimization iterations for a typical (*left panel*) and an extreme (*right panel*) observation $y$](https://anonymous.4open.science/r/CompDoobTransform/figures/OU_value_train.pdf)
>
> OU-C. [Evolution of neural network $-\mathrm{N}(x,y,t)$ (*black to copper*) approximating the control function $c_{\star}(x,y,t)$ (*red*) over first $500$ optimization iterations for a typical (*upper row*) and an extreme (*lower row*) observation $y$](https://anonymous.4open.science/r/CompDoobTransform/figures/OU_control_train.pdf)
>
> **Ornstein-Uhlenbeck model after training**
>
> OU-D. [Evolution of loss estimate over $2000$ optimization iterations under static (*red*) and iterative (*blue*) CDT schemes](https://anonymous.4open.science/r/CompDoobTransform/figures/OU_loss_compare.pdf)
>
> OU-E. [Neural network approximation $\mathrm{N}_0(x,y)$ of the initial value function $v(x,y,0)$ (*purple*) after training with the static (*red*) and iterative (*blue*) CDT schemes for a typical (*left panel*) and an extreme (*right panel*) observation $y$](https://anonymous.4open.science/r/CompDoobTransform/figures/OU_value_compare.pdf)
>
> OU-F. [Neural network approximation $-\mathrm{N}(x,y,t)$ of the control function $c_{\star}(x,y,t)$ (*purple*) after training with the static (*red*) and iterative (*blue*) CDT schemes for a typical (*upper row*) and an extreme (*lower row*) observation $y$](https://anonymous.4open.science/r/CompDoobTransform/figures/OU_control_compare.pdf)
>
> **Logistic diffusion model during initial training phase**
>
> LD-A. [Evolution of loss estimate over first $500$ optimization iterations](https://anonymous.4open.science/r/CompDoobTransform/figures/logistic_loss_train.pdf)
>
> LD-B. [Evolution of neural network $\mathrm{N}_0(x,y)$ (*black to copper*) approximating the initial value function $v(x,y,0)$ over first $500$ optimization iterations for a typical (*left panel*) and an extreme (*right panel*) observation $y$](https://anonymous.4open.science/r/CompDoobTransform/figures/logistic_value_train.pdf)
>
> LD-C. [Evolution of neural network $-\mathrm{N}(x,y,t)$ (*black to copper*) approximating the control function $c_{\star}(x,y,t)$ over first $500$ optimization iterations for a typical (*upper row*) and an extreme (*lower row*) observation $y$](https://anonymous.4open.science/r/CompDoobTransform/figures/logistic_control_train.pdf)

---

> ### Author Response · Authors · 2022-08-02
> **Response to Reviewer qRy1 [3 / 4]**
>
> **Logistic diffusion model after training**
>
> LD-D. [Evolution of loss estimate over $2000$ optimization iterations under static (*red*) and iterative (*blue*) CDT schemes and various levels of informative observations](https://anonymous.4open.science/r/CompDoobTransform/figures/logistic_loss.pdf)
>
> LD-E. [Neural network approximation $\mathrm{N}_0(x,y)$ of the initial value function $v(x,y,0)$ after training with the static (*red*) and iterative (*blue*) CDT schemes for a typical (*left panel*) and an extreme (*right panel*) observation $y$](https://anonymous.4open.science/r/CompDoobTransform/figures/logistic_value_compare.pdf)
>
> LD-F. [Neural network approximation $-\mathrm{N}(x,y,t)$ of the control function $c_{\star}(x,y,t)$ after training with the static (*red*) and iterative (*blue*) CDT schemes for a typical (*upper row*) and an extreme (*lower row*) observation $y$](https://anonymous.4open.science/r/CompDoobTransform/figures/logistic_control_compare.pdf)
>
> **Cell model after training**
>
> Cell-A. [Evolution of loss estimate over $2000$ optimization iterations under static (*red*) and iterative (*blue*) CDT schemes and various levels of informative observations](https://anonymous.4open.science/r/CompDoobTransform/figures/cell_loss.pdf)
>
> Cell-B. [Neural network approximation $\mathrm{N}_0(x,y)$ of the initial value function $v(x,y,0)$ after training with the static (*left column*) and iterative (*right column*) CDT schemes for a typical (*upper row*) and an extreme (*lower row*) observation $y$](https://anonymous.4open.science/r/CompDoobTransform/figures/cell_value_function.pdf)
>
> Cell-C. [Neural network approximation $-\mathrm{N}(x,y,t)$ of the control function $c_{\star}(x,y,t)$ after training with the static (*upper row*) and iterative (*lower row*) CDT schemes for a typical observation $y$](https://anonymous.4open.science/r/CompDoobTransform/figures/cell_control_function_in.pdf)
>
> Cell-D. [Neural network approximation $-\mathrm{N}(x,y,t)$ of the control function $c_{\star}(x,y,t)$ after training with the static (*upper row*) and iterative (*lower row*) CDT schemes for an extreme observation $y$](https://anonymous.4open.science/r/CompDoobTransform/figures/cell_control_function_out.pdf)
> ***
> 8. > Could you be more precise on which variables you are taking expectations over and why? This is important for the loss.
>
> We described in Line 157 that: "The above expectation is with respect to the distribution of the Brownian motion $B_t$, the initial condition $X_0^{c,Y}\sim \eta_X$ of the controlled diffusion, and the observation $Y\sim \eta_Y$ at time T." We hope that this description is clear enough, and would be happy to add more clarification in the text if the reviewer thinks it is necessary.
> ***
> 9. > CDT algorithm, it would help to have a more detailed pseudo code for the algorithm.
>
> This point is well-noted. In the updated version of the manuscript, we expand on the algorithmic description (contained in between Lines 169-180) by making references to some algorithmic functions which we will detail in the appendix. These functions will provide more detailed description of each step of the CDT algorithm using PyTorch-like pseudo-code.
> ***
> 10. > Doob’s transform: I suggest you either give more details or remove the sketch from the main paper. As is it is not useful.
>
> As Doob's $h$-transform is a concept from applied probability, most exposition of it in existing literature tend to be quite technical in nature due to the mathematical formalism involved. Our aim in Sections 2.3 and 2.4 is to provide an intuitive and instructive introduction of Doob's $h$-transform by dispensing with technical complications. That said, as our exposition relies on a few concepts in SDE theory, we acknowledge that it might still be inaccessible for some readers. We have added some overarching discussions in these two sections to help such readers get the key takeaways needed to understand the rest of our article.
> ***

---

> ### Author Response · Authors · 2022-08-02
> **Response to Reviewer qRy1 [4 / 4]**
>
> 11. > Where is the conclusion? It is good to end a paper with the conclusion summarizing the method and results.
>
> This remark has also been raised by other reviewers. We have added a conclusion that recapitulates the proposed method and numerical results, and also includes the following discussion points.
>
> A. Contrarily to a number of methods in existing literature, the proposed method does not address the SDE time-discretization bias. One could consider employing randomization strategies developed by [RG2015, JLY2020] to remove this bias.
>
> B. Our proposed method is widely applicable as it does not rely on any specific structure of the diffusion. Furthermore, it can yield consistent estimates of the filtering distributions as both the number of discretization steps and the number of particles go to infinity.
>
> C. It is to be contrasted with a number of (very useful) methods proposed in the data-assimilation and geoscience literature that rely on Gaussianity assumptions or model-specific structure. These methods, typically designed to operate in very high-dimensional settings, achieve lower variance by increasing the bias. Future work will explore how to tailor Doob's $h$-transform for problems in geoscience by employing techniques such as localization [P2016].
>
> D. Future work will also experiment with alternative formulations and  parameterizations of the neural networks approximating the initial value and control functions to accelerate the training procedure [CWN2019, NR2021].
> ***
> We hope we have managed to address your concerns about our work, and would be grateful if you could update your evaluation of our submission accordingly. Please feel free to leave us any additional comments or suggestions.
> ***
> [CWN2019] Chan-Wai-Nam Q, Mikael J, Warin X. Machine learning for semi linear PDEs. Journal of scientific computing. 2019 Jun;79(3):1667-712.
>
> [JLY2020] Jasra A, Law KJ, Yu F. Unbiased filtering of a class of partially observed diffusions. Advances in Applied Probability. 2020:1-27.
>
> [NR2021] Nüsken N, Richter L. Solving high-dimensional Hamilton–Jacobi–Bellman PDEs using neural networks: perspectives from the theory of controlled diffusions and measures on path space. Partial Differential Equations and Applications. 2021 Aug;2(4):1-48.
>
> [P2016] Poterjoy, J. (2016) A localized particle filter for high-dimensional non- linear systems. Monthly Weather Review, 144, 59–76. https://doi.org/ 10.1175/MWR-D-15-0163.1.
>
> [RG2015] Rhee CH, Glynn PW. Unbiased estimation with square root convergence for SDE models. Operations Research. 2015 Oct;63(5):1026-43.

---

> ### Comment · Reviewer_qRy1 · 2022-08-08
> **Thanks for the thorough comments**
>
> Thank you for the thorough comments that clarify my understanding very positively.
> For some reason I currently can't access the additional documents on the anonymized repo (I will try again)
> I m confident the authors can implement the proposed changes to the manuscript within short delays.
> I have no further questions.

---

> > ### Author Response · Authors · 2022-08-08
> > **Response concerning additional figures**
> >
> > Thank you for going through our reply and for your kind response. Unfortunately, we are also having trouble accessing our additional figures on Anonymous Github at the moment, as its server seems to be down. We will monitor the situation and provide an alternative link if this issue persists. As these figures are an integral part of our revision, we sincerely hope the reviewer will take these figures into account in the evaluation of our paper.

---

> > > ### Author Response · Authors · 2022-08-09
> > > **Update concerning additional figures**
> > >
> > > Here is an update. We have just gotten confirmation that the issue with the Anonymous Github server has been fixed. The links to all additional figures are now working properly. Thank you for bringing this issue to our attention.

---

### Author Response · Authors · 2022-08-02
**Response to all reviewers**

We would like to thank all reviewers for your time and feedback. We appreciate the thoughtful comments and positive response.

We have provided detailed responses to each reviewer. The following is a summary of (what we think are) the key points from the collated feedback:

1. provide more motivation for why online filtering of diffusions is an important problem with a wide range of applications;

2. better position our contribution with respect to related work;

3. provide more intuition in the background section;

4. give more insight on how our neural network approximations evolve during training, how fast training iterations converge, and visualize the trained neural networks;

5. illustrate the applicability of our approach in higher-dimensional settings where it is computationally infeasible to consider classical PDE solvers;

6. clarify whether alternative particle filters are applicable in this problem setting and compare our proposed auxiliary particle filter to not just the bootstrap particle filter but also the current state-of-the-art;

7. add a conclusion recapitulating the main points of our approach and results, and a discussion of future work.

This is admittedly a long list and we have done a major revision to thoroughly address each point. Please feel free to leave us any additional comments or suggestions.

---

### Meta-Review · Area_Chair_AsFf · 2022-08-26

**Recommendation:** Reject
**Confidence:** Less certain

**Metareview:**

This is an interesting paper, but it also requires some additional work. The paper received mixed responses from the reviewers, but the main concerns were sorted out during the rebuttal and discussion. In the end, the reviewers ended up (weakly) recommending acceptance after the authors promised an extensive number of changes to the paper in the camera-ready stage. However, the promised changes appear very extensive and beyond what would be expected for a typical NeurIPS paper (list below). The authors have provided links to plots in the discussion, but have not revised the paper in OpenReview. Sorting out the issues counts as a 'major revision' and the conference review process does not recognise such a concept.

List of changes (based on rebuttals):
* Empirical comparison (& discussion) to GIRF instead of only bootstrap particle filter
* Adding a discussion section to the end of the paper
* Writing a more thorough Related work section
* A study on the impact of growing dimensionality (OU process)
* Plots and discussion on the learned dynamics (control)
* Plots and discussion on training loss
* A more intuitive explanation of the Doob's h-transform in the Background section
* Expanded Introduction to discuss applications (with references)
* Computational cost discussion (promised to add to the appendix)
* Figures illustrating Doob’s h-transform trajectories
* Discussion on how fine discretization is needed
* Discussion on parameter inference and the smoothing problem
* Discussion on irregular observation intervals

**Award:**

No

---

### Decision · Program_Chairs · 2022-09-14

Reject